# Isotopic measurements in water vapor, precipitation, and seawater during EUREC⁴A

Adriana Bailey[1], Franziska Aemisegger[2], Leonie Villiger[2], Sebastian A. Los[3], Gilles Reverdin[4], Estefanía Quiñones Meléndez[5], Claudia Acquistapace[6], Dariusz B. Baranowski[7], Tobias Böck[6], Sandrine Bony[8], Tobias Bordsdorff[9], Derek Coffman[10], Simon P. de Szoeke[5], Christopher J. Diekmann[11,12], Marina Dütsch[13,14], Benjamin Ertl[11,15], Joseph Galewsky[3], Dean Henze[5], Przemyslaw Makuch[16], David Noone[17,5], Patricia K. Quinn[10], Michael Rösch[18], Andreas Schneider[9,19], Matthias Schneider[11], Sabrina Speich[20], Bjorn Stevens[21], Elizabeth J. Thompson[22]

[1]National Center for Atmospheric Research, Boulder, CO, USA
[2]Institute for Atmospheric and Climate Science, ETH Zurich, Zurich, Switzerland
[3]Department of Earth & Planetary Sciences, University of New Mexico, Albuquerque, NM, USA
[4]Laboratoire d'Océanographie et du Climat: expérimentation et approches numériques (LOCEAN/IPSL), Sorbonne Université-CNRS-IRD-MNHN, Paris, France
[5]College of Earth, Ocean, and Atmospheric Sciences, Oregon State University, Corvallis, OR, USA
[6]Institute for Geophysics and Meteorology, University of Cologne, Köln, Germany
[7]Institute of Geophysics Polish Academy of Sciences, Warsaw, Poland
[8]Laboratoire de Météorologie Dynamique (LMD/IPSL), CNRS, Sorbonne University, Paris, France
[9]Netherlands Institute for Space Research, SRON, Leiden, the Netherlands
[10]NOAA Pacific Marine Environmental Laboratory (PMEL), Seattle, WA, USA
[11]Institute of Meteorology and Climate Research (IMK-ASF), Karlsruhe Institute of Technology (KIT), Karlsruhe, Germany
[12]now at Telespazio Germany GmbH, Darmstadt, Germany
[13]University of Vienna, Vienna, Austria
[14] University of Washington, Seattle, WA, USA
[15]Steinbuch Centre for Computing (SCC), Karlsruhe Institute of Technology (KIT), Karlsruhe, Germany
[16]Institute of Oceanology Polish Academy of Sciences, Sopot, Poland
[17]Department of Physics, University of Auckland, Auckland, New Zealand
[18]Department of Environmental Systems Science, ETH Zurich, Zurich, Switzerland
[19]Finnish Meteorological Institute, Sodankylä, Finland
[20]LMD/IPSL, École Normale Supérieure, CNRS, Paris, France
[21]Max Planck Institute for Meteorology, Hamburg, Germany
[22]NOAA Physical Sciences Laboratory, Boulder, CO, USA

*Correspondence to:* Adriana Bailey (abailey@ucar.edu)

**Abstract.** In early 2020, an international team set out to investigate trade-wind cumulus clouds and their coupling to the large-scale circulation through the field campaign EUREC⁴A: ElUcidating the RolE of Clouds-Circulation Coupling in ClimAte. Focused on the western tropical Atlantic near Barbados, EUREC⁴A deployed a number of innovative observational strategies, including a large network of water isotopic measurements collectively known as EUREC⁴A-iso, to study the tropical shallow convective environment. The goal of the isotopic measurements was to elucidate processes that regulate the hydroclimate state – for example, by identifying moisture sources, quantifying mixing between atmospheric layers, characterizing the microphysics that influence the formation and persistence of clouds and precipitation, and providing an extra constraint in the evaluation of numerical simulations. During the field experiment, researchers deployed seven water vapor isotopic analyzers on two aircraft, on three ships, and at the Barbados Cloud Observatory (BCO). Precipitation was collected for isotopic analysis at the BCO and from

aboard four ships. In addition, three ships collected seawater for isotopic analysis. All told, the in-situ data span the period 5 January through 22 February 2020 and cover the approximate area 6 to 16°N and 50 to 60°W, with water vapor isotope ratios measured from a few meters above sea level to the mid-free troposphere and seawater samples spanning the ocean surface to several kilometers depth.

This paper describes the full EUREC$^4$A isotopic in-situ data collection – providing extensive information about sampling strategies and data uncertainties – and also guides readers to complementary remotely sensed water vapor isotope ratios. All field data have been made publicly available even if they are affected by known biases, as is the case for high-altitude aircraft measurements, one of the two BCO ground-based water vapor time series, and select rain and seawater samples from the ships. Publication of these data reflects a desire to promote dialogue around

improving water isotope measurement strategies for the future. The remaining, high-quality data create unprecedented opportunities to close water isotopic budgets and evaluate water fluxes and their influence on cloudiness in the trade-wind environment. The full list of dataset DOIs and notes on data quality flags are provided in Table 3, Sect. 5 Data Availability.

## 1 Introduction

In an effort to solve unanswered questions about tropical low-level clouds and their sensitivity to the larger trade-wind marine environment, researchers in early 2020 carried out a multi-national, multi-platform field campaign called EUREC$^4$A – ElUcidating the RolE of Clouds-Circulation Coupling in ClimAte (Bony et al., 2017; Stevens et al., 2021). EUREC$^4$A took place in the tropical western Atlantic near the island nation of Barbados and deployed an innovative array of measurement platforms, which included multiple research aircraft and ships, aerial and oceanic

drones, and ground-based stations. EUREC$^4$A was comprised of various research components. Some, like ATOMIC (Atlantic Tradewind Ocean-Atmosphere Mesoscale Interaction Campaign; Quinn et al., 2021; Pincus et al., 2021) and EUREC$^4$A-OA (EUREC$^4$A Ocean Atmosphere interactions; http://eurec4a-oa.eu), were formally coordinated efforts that supported the deployment of the large research facilities. Others, like EUREC$^4$A-iso – the focus of this paper, were informally coordinated by individual investigators.

     EUREC$^4$A-iso supported EUREC$^4$A's investigations of moist processes and their effects on trade-wind cloudiness through the deployment of an expansive network of isotopic measurements in atmospheric water vapor, seawater, and precipitation. EUREC$^4$A-iso also contributed its own set of complementary research objectives to the larger EUREC$^4$A mission. These were to

● link isotopic signals to patterns of cloud organization,
     ● refine estimates of the sub-cloud and cloud layer isotopic budgets,
     ● determine the importance of large-scale advection in influencing these layers,
     ● evaluate the roles of ocean fluxes and rain evaporation in moistening the atmosphere within cold pools, and
     ● characterize the coupling between tropical and extratropical water cycles.

Efforts to evaluate these objectives are leveraging the in-situ data in conjunction with remotely sensed water vapor isotope ratios and numerical modeling experiments, ranging from global to large-eddy simulations. The measurements collected in 2020 thus lay the foundation for in depth scientific investigations that combine measurements and models at distinct scales to tackle open questions about the water cycle in trade-wind regions.

Because isotope ratios (i.e. $^{18}O/^{16}O$, D/H) are sensitive to the integral of moist processes experienced by an air mass during transport (Gat 1996; Galewsky et al., 2016), they are an ideal tool for assessing the coupling between the circulation at large scales and moist processes at smaller scales. This sensitivity stems from the fact that isotopically heavy and light water molecules change phase and diffuse at distinct rates, causing the heavier molecules to reside in greater relative abundance in the condensed phase. The result is that the atmosphere is depleted of heavy isotopes

relative to ocean water below and becomes further depleted as condensation and rainout occur. In contrast,

evaporation from the ocean, and subsequent upward moisture transport, enriches the atmosphere isotopically (even though the evaporative process itself discriminates against heavy water). Evaporation also causes a shift in the hydrogen isotope ratio relative to the oxygen isotope ratio due to diffusive differences between the heavy isotopologues ($H_2^{18}O$ and HDO) under non-equilibrium conditions. Isotope ratios can thus help differentiate

between air masses that have experienced distinct water cycle histories and test hypotheses about the processes responsible for setting air mass humidity and cloud states – processes such as ocean evaporation, precipitation development, rain evaporation, turbulent mixing, and large-scale advection (Fig. 1; Worden et al., 2007; Noone et al., 2011; Hurley et al., 2012; Bailey et al., 2013; Benetti et al., 2015; Aemisegger et al., 2015; Salmon et al., 2019; Risi et al., 2020; Aemisegger et al., 2021a).


While the application of water isotopes to the study of modern hydroclimate processes has been steadily growing, EUREC[4]A stood out from past efforts through its successful coordination of isotopic measurements across multiple platforms and through the sheer quantity of isotopic data it collected. A total of seven water vapor isotopic analyzers, sampling at 0.5 Hz or faster, were deployed during the campaign: two on research aircraft, three on

ocean-going research vessels, and two collocated at the long-term cloud observatory that operates on the eastern shores of Barbados (Stevens et al., 2016). These continuous measurements – which have been processed at 1 s to 2 minute time resolution – were complemented by sampling of precipitation collected both onshore and on ships and seawater at various depths. This unique collection of data meets a growing need for isotopic measurements that can help close water budgets and provide sufficient spatiotemporal coverage to evaluate simulations of moist processes

meaningfully. Moreover, all isotopic measurements were integrated alongside numerous other meteorological and oceanographic measurements, many of which were designed to characterize large-scale vertical motions, convective mass fluxes, cloud micro- and macrophysics, and air-sea exchange. This wealth of observational data will aid interpretation of the isotopic signals, just as the isotopic information will provide an additional lens through which to evaluate microphysical and dynamical controls on trade-wind cloudiness.


This paper describes the collective EUREC[4]A-iso in-situ dataset and provides detailed information about the isotopic measurement systems deployed, the ways in which data were processed, the measurement uncertainties, and data formatting and distribution. Particular attention has been paid to quantifying and reporting uncertainties, especially for the isotopic measurements in water vapor. Uncertainties in these data reflect the diverse operating

conditions and constraints associated with each platform and the need to tailor post-processing corrections to individual instrument performance, as is considered best practice (Aemisegger et al., 2012; Bailey et al., 2015). We expect that these reporting efforts will not only promote more accurate cross-platform comparisons but also raise the bar for characterizing uncertainties in field-based water vapor isotopic deployments—which have grown in number over the last decade (Galewsky et al., 2016).


Extensive efforts have also been made to flag, rather than mask, data whose quality concerns cannot be described by error estimates. These data include high-altitude (free tropospheric) oxygen isotopic measurements from the aircraft, one water vapor isotopic time series from Barbados, rain samples whose collection times were delayed by multiple hours or days, and a group of seawater samples whose storage caps "breathed" during transit. Though many of these

samples are not recommended for scientific analysis, they are included in the published datasets so that lessons can be learned and applied to improve instrument performance, sampling installation, and collection and calibration protocols. Such improvements will help promote regular integration of water isotopic measurements in future large-scale field campaigns and regular comparison of water vapor isotopic data with model output. A summary of all available data and notes on flagged data are provided in Table 3 of Section 5.


Finally, in addition to describing the in-situ data, this paper also guides readers to complementary datasets, including EUREC[4]A-iso remotely sensed isotope ratios. The satellite-based data have been repackaged into custom subsets that cover an extensive region surrounding Barbados for the period in which the 2020 field campaign occurred. Even though the remote sensors exhibit vastly different sensitivities to the atmosphere than the water vapor isotopic

analyzers deployed during EUREC[4]A, they provide large-scale spatial and temporal context for the in-situ data, which enhances our ability to investigate shallow convective cloud regimes and their role in climate.

## 2 Data collection

During EUREC[4]A, seven laser-based analyzers measured the concentration of water vapor and its isotopic composition from ground-based, airborne, and ship-based platforms. Two analyzers made collocated ground-based
measurements at the Barbados Cloud Observatory (hereafter BCO, https://barbados.mpimet.mpg.de/), airborne vapor measurements were made aboard the French ATR-42 (hereafter ATR) operated by SAFIRE (Bony et al., 2022) and the US NOAA WP-3D Orion (hereafter P-3; Pincus et al., 2021), and ship-based vapor measurements were made aboard the French research vessel *L'Atalante* (hereafter Atalante), the German research vessel *Meteor* (hereafter Meteor), and the US NOAA research vessel *Ronald H. Brow*n (hereafter Brown; Quinn et al., 2021).

The seven deployed analyzers all measure water vapor concentration as a mole fraction relative to (total) moist air (i.e. $n_{wv}/n_t$) and report the isotope ratios of oxygen and hydrogen in water vapor in delta notation (units ‰):

$$\delta^{18}O = ([^{18}O/^{16}O]_{obs}/[^{18}O/^{16}O]_{VSMOW} - 1) \times 1000, \qquad\qquad (1)$$

$$\delta D = ([D/H]_{obs}/[D/H]_{VSMOW} - 1) \times 1000, \qquad\qquad (2)$$

where *obs* indicates observed, and *VSMOW* is Vienna Standard Mean Ocean Water (IAEA, 2017).

In addition, precipitation samples for isotopic analysis were collected from the BCO, from the three aforementioned vessels, and from the German research vessel *Maria S. Merian* (hereafter Merian). Seawater samples for isotopic analysis were also collected by the Meteor, the Brown, and the Atalante during their research cruises. Figure 2 shows representative sampling locations of the various isotopic measurement platforms, and Fig. 3 illustrates the time periods of continuous and discrete sampling. Further details of the measurement systems, the in-field
calibration protocols, and the rain and seawater collection procedures are described for each type of observational platform below. Any pre- and post-campaign calibrations used in post-processing the water vapor concentration and isotope ratio data are discussed in Sect. 3.

### 2.1 Ground-based isotopic measurements

Ground-based isotopic measurements were set up at the BCO, which served as the central land-based observatory
during EUREC[4]A. Operated by the Max Planck Institute for Meteorology together with the Caribbean Institute for Meteorology and Hydrology, the BCO is situated on a promontory 17 m a.s.l. at Deebles Point (13.16°N, 59.43°W), near the most windward point of the island of Barbados. As a result, the BCO is directly exposed to the North Atlantic trade winds, and no island effects on the flow or atmospheric water budget have been detected previously (Stevens et al., 2016). Moreover, clouds observed at the BCO are typical of trade-wind clouds across the tropics
(Medeiros and Nuijens, 2016), making the observatory well-situated for investigating shallow convective processes that are regionally representative of the tropical trade-wind environment.

### 2.1.1 Water vapor isotopic measurements at the BCO

The BCO water vapor isotopic measurements were designed to serve as a continuous fixed-point dataset with 1 minute time resolution at a location with extensive meteorological in-situ and remote sensing observations,
including continuous Raman lidar water vapor profiling and passive remote sensing of column water vapor and condensed water. Vapor isotopic measurements at the BCO were made with two laser spectrometric instruments

installed side-by-side (cf. Aemisegger et al. 2021): a Picarro L1115 analyzer, using cavity ring-down spectroscopy (CRDS), and a Los Gatos Research (LGR) analyzer, using off-axis integrated cavity output spectroscopy (OA-ICOS). The two systems operate at different wavelengths in the infrared; consequently, baseline effects due to varying water vapor concentrations can affect the measurements differently (Johnson and Rella, 2017).

The two instruments were installed in tandem to permit cross-validation and ensure a high-quality ground-based time series for the duration of the EUREC[4]A field deployment. Moreover, because the two analyzers did not sample reference gas simultaneously, the possibility exists of gap-filling the ambient time series of one analyzer with the ambient time series of the other. The CRDS system measured from 13 January through 17 February 2020 (DOY 13-48), while the OA-ICOS system operated over a slightly shorter period from 18 January to 16 February 2020 (DOY 18-47; Fig. 3).

As shown in Fig. 4, the laser spectrometers were installed in a temperature-regulated container ($24\pm2°C$). A KNF pump was used to pull ambient air into the container through an 8.5 m long, heated (12 mm OD) PTFE line at a flow rate of 15 L min$^{-1}$. The inlet was hooked downward near the intake and shielded from rainfall and sea spray by a funnel. The two spectrometers picked off sub-samples of the main flow through a narrower 0.3 m long isolated PTFE tube (Fig. S1). This configuration resulted in a sample residence time of just 3 s within the inlet line. However, because residence times within the instruments are much longer (e.g. 60 s for the CRDS system), total measurement response times are closer to 63 s.

Calibration checks were performed in the field by sampling reference gas generated from three liquid standards that spanned [-20.96, 4.52]‰ for $\delta^{18}O$ and [-171.01, 17.70]‰ for $\delta D$ (Supplemental Information). The standards were measured daily for 20-to-60 minutes each. During the first half of the campaign (13 January to 2 February 2020, DOY 13-33), an LGR water vapor isotope standard source (WVISS) was used for producing reference gas from the liquid standards for both spectrometers. Due to an unfortunate breakdown of the WVISS system, a LI-COR dew point generator (LI-610) was used thereafter (until 17 February 2020, DOY 48).

### 2.1.2 Precipitation isotopic measurements at the BCO

To complement the water vapor isotopic measurements and provide an opportunity to evaluate exchange processes between precipitation and the environment, rainwater samples were also collected at the BCO. Samples were collected on an event basis, as well as at higher resolution (every 10 minutes) during a targeted intensive observation period of a trailing cold front on 22 January 2020 (DOY 22) (Fig. 3; see Villiger et al., 2022 for a detailed overview of the event). In total, 42 rain samples were obtained between 16 January and 18 February 2020 (DOY 16-49), 26 of which were from the cold front passage.

Rainwater was collected using the Palmex RS1 precipitation sampling system, which is specially designed to avoid post-sampling evaporation (Gröning et al., 2012) and has been used by the International Atomic Energy Agency (IAEA) for its Global Network of Isotopes in Precipitation (GNIP). The sampler consists of a mesh-guarded funnel, which drains rainwater through a 4 mm ID tube into a 3 L PVC collection bottle. At the BCO, the sampler was installed unshielded on the top of the 2.6 m high container in which the vapor isotopic analyzers were housed (Fig. 4).

Samples were collected as soon as possible following a precipitation event, and the amount of rain collected was weighed with a portable scale. Rainwater from the PVC collection bottle was then transferred into 1.5 mL analysis vials, which were filled to the brim and sealed with parafilm. Except when in transit, the sealed samples were stored at room temperature (20-25°C) before analysis at the University of Freiburg after the field experiment.

## 2.2 Airborne isotopic measurements

Airborne measurements were a key component of EUREC[4]A, providing an intimate look at the shallow convective systems and clouds targeted by the campaign (Stevens et al., 2021). A total of four crewed aircraft participated in EUREC[4]A, two of which – the ATR and the P-3 – carried water vapor isotopic analyzers on board.

### 2.2.1 ATR water vapor isotopic measurements

The ATR flew predominantly at cloud base and in the sub-cloud layer on the eastern side of the so-called HALO circle – a 200 km diameter upper altitude circle, centered at 13.30° N, 57.72° W, approximately 150 km to the east of Barbados. The circle was so named because the German Aerospace Center's (DLR) HALO aircraft (Konow et al., 2021) launched dropsondes around the circle to estimate large-scale vertical motions (Bony et al., 2017; Stevens et al., 2021). The ATR spent most of its flight hours flying repeated rectangles and L-legs to characterize the shallow cumulus field and to measure boundary-layer properties; it also flew occasionally at cloud top and sampled the lower free troposphere during ferry legs (Bony et al., 2022). The ATR was equipped with various remote-sensing (lidar, radar) and in-situ (turbulence, radiation, microphysics, stable water isotopes) instruments (Chazette et al., 2020; Brilouet et al., 2021). ATR flights were closely coordinated with HALO flights and typically lasted 4-5 hours in duration, thus making it possible to conduct two flights per day. In total, 19 flights were conducted between 25 January 2020 and 13 February 2020 (DOY 25-44).

Isotopic measurements aboard the ATR were made with a customized, fast-response version of Picarro's L2130-i cavity ring-down spectrometer (with nominal sampling frequency of 1 Hz). The analyzer had been deployed in previous field campaigns, both for near-surface (Thurnherr et al., 2020) and airborne (Sodemann et al., 2017) measurements. To selectively target atmospheric moisture in vapor phase, the analyzer was installed behind a rearward-facing gooseneck inlet constructed from a 0.3 m length of 1/4 inch OD stainless steel tubing that was mounted to the starboard side of the aircraft fuselage (Fig. 5a). Ambient air was pumped at a rate of 13 SLPM (with reference temperature 20°C and reference pressure 1013.25 hPa) through the gooseneck, past a particle filter, and down a 1.5 m long, 10 mm ID PTFE tube heated to 80°C (Fig. S2). The analyzer picked off a sub-sample of the flow at a rate of 0.28 SLPM through a thermally isolated, 0.2 m long, ¼ inch OD PTFE line. The resulting time delay from the intake to the analyzer was just 1 s; however, additional delays within the instrument itself resulted in a total measurement response time of 10 s (at sea level).

Calibration checks were performed using a Picarro Standards Delivery Module (SDM), which was installed on the aircraft instrument rack. The SDM delivers a thin stream of liquid water of known isotopic composition into a vaporizer, which, in turn, converts the stream to gas phase. Each flight day, 2-3 liquid standards, with values spanning [-55.79, -11.13]‰ for $\delta^{18}O$ and [-439.71,-80.42]‰ for $\delta D$, were run for 20 minutes each either before or after the day's flights (Supplemental Information). When time permitted, the SDM was also run during the midday refueling. Four additional calibration checks were performed in flight for a total of 32 checks over the course of the campaign. Aliquots of the liquid standards were also taken regularly during the campaign to survey any potential drift in the standards themselves.

### 2.2.2 P-3 water vapor isotopic measurements

Compared to the ATR, the P-3 flew over a larger altitude range and traversed a wider geographic area, typically east, and frequently upwind, of the HALO circle (Pincus et al., 2021). Most flights, which were about 8 hours in duration, included a circle in the mid-troposphere to launch dropsondes, a rake pattern to deploy Airborne EXpendable BathyThermographs (AXBTs), and vertically stacked level legs, ascending from 150 m.a.s.l, through cloud, to cloud top. Particularly relevant for water isotopic measurements, there were also continuous slantwise ascents and descents, some spanning approximately 150 to 7800 m. Together with the stacked cloud passes and

takeoffs and landings, these slantwise traverses provide several snapshots per flight of the isotopic profile of the convective environment. Eleven total flights were conducted over the period 17 January to 11 February 2020 (DOY 17-42).

Like the ATR, the P-3 flew a customized, fast-response version of Picarro's L2130-i cavity ring-down spectrometer (with 5 Hz nominal sampling frequency). The analyzer was installed behind a rearward-facing National Center for Atmospheric Research HIAPER Modular Inlet (NCAR HIMIL) that was mounted to the starboard side of the aircraft fuselage (Fig. 5b). Ambient air was pulled through the HIMIL to the analyzer via a 2.1 m long, ¼ inch OD copper tube that was heated to 40°C during the first two flights and 48°C thereafter, once the heat controller's precision was confirmed. (No change in measurement accuracy was detected as a result). As described in Pincus et al. (2021), mass flow through the copper tube was controlled by the spectrometer at 0.63 SLPM, resulting in typical time delays through the inlet line of $(3.4\pm0.3)$ s near sea level and 1.4 s at the highest flight altitudes near 400 hPa.

Calibration checks were performed during three non-flight days by manually injecting liquid standards at least 5 times each directly into a Picarro vaporizer that was temporarily attached to the analyzer for this purpose. A rotation of five liquid standards was used, with isotopic values spanning [-45.41, -0.28]‰ for $\delta^{18}O$ and [-355.18, 1.60]‰ for $\delta D$. However, as described in the Supplemental Information, these in-field calibration measurements were only used to evaluate uncertainty rather than to calibrate the isotopic measurements. The analyzer also sampled from a rack-mounted LI-COR 610 dew point generator usually twice per flight for approximately 10 minutes each time, yet the reference gas generated by the LI-COR proved isotopically unstable over the course of the campaign and was therefore not used to evaluate instrument performance.

### 2.3 Ship-based isotopic measurements

The four ships that participated in EUREC[4]A each sampled water isotope ratios in some form and covered two principal geographic regions: "*Tradewind Alley*", the name given to a corridor stretching approximately eastward from Barbados to the Northwest Tropical Atlantic Station (NTAS), an air-sea flux measuring buoy station near 15°N, 51°W, and the *"Boulevard des Tourbillons"*, a name given to the North Brazil Current eddy corridor along the northern coast of South America (Quinn et al., 2021; Stephan et al., 2021; Stevens et al., 2021). The Meteor and Brown stayed close to Tradewind Alley, providing a valuable ground-up perspective for the EUREC[4]A aircraft flying overhead. In comparison, the Merian and Atalante sailed farther south to observe the atmospheric and oceanic variability near the North Brazil Current and to investigate mesoscale ocean eddies, freshwater inputs from the Amazon and Orinoco discharges, and deep convective outflows from the Intertropical Convergence Zone. The Merian also collected a number of rain samples within Tradewind Alley. Examples of water vapor and precipitation isotopic sampling installations aboard the ships are shown in Fig. 6.

### 2.3.1 Water vapor isotopic measurements at sea

**Meteor**

To provide surface flux and surface-based remote-sensing measurements within Tradewind Alley, the Meteor sampled regularly along a north-south transect defined by the 57.24°W meridian within the eastern portion of the HALO circle. Exceptions to this were, 0400-1100 UTC on 18 January (DOY 18) and 0900-1300 UTC on 19 February (DOY 50) when the ship was stationed about 2 km upwind of the BCO, as well as 1300-2300 UTC on 19 February (DOY 50) when the ship was stationed just offshore of Barbados' Bridgetown port. To sample the isotopic composition of near-surface water vapor, the Meteor operated a 1 Hz CRDS-based Picarro L2130-i analyzer from 18 January until 22 February 2020 (DOY 18-53).

The analyzer aboard the Meteor was housed in the Air-Chemistry Laboratory, the highest enclosed, temperature-
controlled deck on the ship. The analyzer sampled ambient air ~20.3 m.a.s.l. from an inlet affixed to a bow-facing
railing above the Air-Chemistry Laboratory and below the main meteorological instrument mast. The inlet was
composed of a 5 m long, 4.6 mm ID PTFE line, heated to 45ºC and insulated with polyethylene foam and foil tape.
The line's intake was housed in a downward-facing funnel to limit contamination by rainwater and sea spray. The
line also included a 0.2 μm PTFE aerosol filter to limit particle debris. Flow through the line was controlled by the
CRDS system at an approximate rate of 0.03 SLPM, resulting in an expected time delay from the intake to the
analyzer of >2 minutes; however, empirical time-response tests suggest the analyzer could detect initial signal
changes after just 37 s.

Calibration checks were performed daily during the cruise using a rotation of four liquid water standards whose
values spanned [-20.97, -2.79]‰ for $\delta^{18}O$ and [-158.13, -13.12]‰ for $\delta D$ (Table S1). The standards were delivered
to the analyzer in gas phase using a Picarro SDM and vaporizer. Each day, two standards were measured for 10
minutes each, and a new standard was swapped in every four days to complete the rotation.

### Brown

The Brown, like the Meteor, ventured predominantly within Tradewind Alley but tended to sample farther to the
east – as far as 51°W – to provide information about the atmosphere-ocean system upwind of the primary EUREC[4]A
study region (Quinn et al., 2021). The Brown was stationed in port at Bridgetown, Barbados from 1215 UTC on 26
January until 2215 UTC on 28 January (DOY 26-28) and from 1900 UTC on 4 February until 1600 UTC on 6
February (DOY 35-37). To measure the isotopic composition of near-surface water vapor, the Brown operated a
customized 5 Hz Picarro L2130-i analyzer from 26 January to 10 February 2020 (DOY 26-41).

The analyzer aboard the Brown was housed within a measurement container alongside aerosol instrumentation on
the O2 deck of the ship, two levels above the main deck (Quinn et al., 2021). All instruments within the container
sampled from a heated mast whose cone-shaped nozzle was mounted 18 m.a.s.l. (Bates et al., 2002). Air was
pumped through the nozzle and down the 0.2 m diameter mast at a rate of 1000 L min⁻¹. The isotopic analyzer drew
a sub-sample of air from the base of the mast, at a rate of 0.43 SLPM, through a ¼ inch OD, 3 m long copper tube
heated to 50°C and insulated with polyethylene foam. While the time delay from the mast nozzle to the analyzer is
not known precisely, our best estimate is that it was 19 s.

Only one calibration check was performed during the campaign on the day the analyzer was installed on the ship (26
January 2020, DOY 26). The check was performed by manually injecting three liquid water standards – with values
spanning [-22.38, -1.89]‰ for $\delta^{18}O$ and [-163.50, -8.37]‰ for $\delta D$ – into a vaporizer attached to the instrument. Each
standard was injected 5-6 times. Complications in retrieving the instrument from the aerosol measurement container
once the EUREC[4]A deployment had concluded prevented a timely post-campaign verification of the instrument
calibration.

### Atalante

The Atalante sailed predominantly to the south of Barbados, contouring the coast of South America to study the
oceanic meso- and submesoscale dynamics of the Boulevard des Tourbillons. From 23 January to 17 February 2020
(DOY 23-48), a 0.5 Hz Picarro L2120-i analyzer operated from an air-conditioned space to the back of the bridge.
Because there was not enough heated line to reach the mast where the ship's main meteorological station was
located, the analyzer sampled through a 10 m long line of 10 mm diameter PFA tubing, heated to 40°C, which was
attached to the railing on top of the bridge, off the starboard side. The upward-facing intake was housed in a
cylindrical cap that shielded the line from rain and sea spray. Ambient air was pumped through the line at a rate of 6
L min⁻¹ resulting in an estimated time delay of about 8 s from the intake to the analyzer.

Calibration checks were performed daily during the cruise, except on 27 and 28 January 2020 (DOY 27-28). A single liquid standard (with $\delta^{18}O$ and $\delta D$ values of -14.95 and -109.7‰, respectively) was delivered to the analyzer in gas phase using an autosampler paired with a Picarro vaporizer. For each daily calibration check, the autosampler injected the standard into the vaporizer 15 times, consuming about 2.5 hours of measurement time each day.

**2.3.2 Precipitation isotopic measurements at sea**

All four EUREC[4]A research vessels sought to collect event-based rainwater samples; however, the regularity of collection varied by platform. Not all precipitation events provided enough rainwater for collection. Moreover, collection times were sometimes delayed significantly past the end of precipitation events. The Meteor, the Atalante, and the Merian used the same Palmex RS1 rain sampler as installed at the BCO (Gröning et al., 2012). The rain sampler on the Brown was slightly different in nature, composed of a large funnel, screwed to a pear-shaped conical
separatory funnel with a stopcock at the bottom (Fig. 6). All samples were isotopically analyzed in established laboratories following the campaign. Additional details about each installation and sampling protocol are described below.

**Meteor**

The Meteor collected a total of 15 samples, representing 15 separate rain events, between 20 January and 19
February (DOY 20-50). Rainwater was collected by a Palmex RS1 rain sampler installed in a relatively unshielded location on the aft, starboard railing of the navigation deck at ~17.5 m.a.s.l. The location was chosen to limit the effects of wind interactions with the ship and to avoid obstruction of the area above the sampler by the main mast. The sampler funnel was cleaned regularly. Immediately after rainfall ended, samples were transferred to 2 mL vials, which were filled to minimize headspace and sealed with parafilm. Rainfall amount was estimated by sample
volume, however, an undercatch of around half was typical compared to the German Weather Service (DWD) rain gauge, designed for ship use, located on the mast. Samples were stored in a refrigerator at ~4ºC during the cruise and again following shipment to the University of New Mexico Center for Stable Isotopes, where they were analyzed.

**Brown**

A total of 12 samples were collected and analyzed from the Brown for the period 5 January till 11 February (DOY 5-42). Not all samples represent distinct events. In some cases, several samples were taken within the same storm. In other cases, rainfall collection times were delayed by up to several days, and samples may represent a weighted average of multiple events.

The Brown's custom rainwater sampler – composed of a large funnel attached to a conical separatory funnel with a stopcock at the bottom – was affixed to the railing on the O3 deck, the third deck above the main deck, off the starboard bow. The sampler was cleaned occasionally to remove sea spray and salt accumulation on the inside walls of the sampler's large funnel. Following a rain event, rainwater was drained from the separatory funnel into 30 mL glass vials that were sealed with PolyCone caps and parafilm. Sample volume was not measured; however, the
optical rain gauge aboard the Brown provides an estimate of precipitation rate (Quinn et al., 2021). Samples were stored at ambient temperature prior to analysis at the University of New Mexico Center for Stable Isotopes.

**Atalante**

A total of seven samples were collected on the Atalante from 20 January to 18 February (DOY 20-49). On eight additional occasions rain was reported on the ship's log but no water was found in the rain sampler, suggesting

winds may have influenced the sampler's collection efficiency. The Palmex RS1 rain sampler used to collect precipitation on the Atalante was affixed to the railing of the upper deck, just below the bridge and on the side towards the prow. Rain was typically collected within an hour of the end of a precipitation event; however, collection times reported in the datafile are not exact. Water height in the rain sampler was observed before collection, providing some sense of rainwater amount. The precise details of sample storage, prior to analysis at the

University of Freiburg, are not known.

**Merian**

The Merian collected a total of 23 rain samples across 16 days between 20 January and 19 February (DOY 20-50). Rain was collected by a Palmex RS1 rain sampler that was affixed to the railing on the "Peildeck" or upper deck of the ship and shielded on the starboard side by the ship's superstructure. Following collection, samples were

transferred to 1.5 and 15 mL vials and sealed with parafilm. Samples were not weighed because rainfall amount was measured directly by a vertically pointing micro rain radar on the ship (MRR; Stephan et al., 2021). Measurements from the radar were also used to attribute an amount-weighted mean rainfall time for each sample, which was then used to identify the vessel's geographic location during precipitation events. Because no rainfall was detected by the radar during one collection period, no geographic location could be assigned. Samples were stored at ambient

temperature before analysis at the University of Freiburg.

**2.3.3 Seawater isotopic samples**

Three ships – the Meteor, Brown, and Atalante – collected seawater for isotopic analysis. All three sampled within 10 m of the ocean surface on a (near) daily basis, and the Brown and Atalante sampled seawater occasionally at greater depths. During three intensive observation periods (Fig. 3), the Meteor and Brown also sampled isotopic

variability in near-surface ocean water across a diel cycle.

**Meteor**

The Meteor collected seawater for isotopic analysis on a near daily basis from a depth of 10 m using the ship's Conductivity, Temperature, and Depth (CTD) profiler. These samples were collected from the CTD cast closest to 1900 UTC. To target a full diel cycle, samples were also collected every two hours during an intensive observation

period (IOP) that took place on 10-11 February 2020 (DOY 41-42) while the Meteor was on station for 24 hours at the northern intersection of its meridional transect with the HALO circle (14.18°N, 57.24°W). A total of 28 daily and 12 IOP seawater samples are available from the Meteor from the period 19 January to 21 February (DOY 19-52).

Following collection, seawater samples were treated with CuCl to prevent isotopic alteration by biotic activity. All samples were then transferred to 2 mL vials without headspace and sealed with parafilm. Samples were stored in a refrigerator at ~4ºC during the cruise and again following shipment to the University of New Mexico Center for Stable Isotopes, where they were analyzed.

**Brown**

Like the Meteor, the Brown sampled both spatial and temporal variability in seawater isotopic composition. From 8 January to 12 February (DOY 8-43) a total of 126 samples were collected by several methods. Forty-four samples were collected by CTD cast across 10 days of the cruise, providing information over a variety of depths. Thirteen surface samples were collected by throwing a bucket overboard from the starboard bow. These samples were taken approximately every six hours, over the course of two 2-day IOPs (Fig. 3), to examine diel variability. And, 69

flowthrough samples were collected from the main ship laboratory to provide a survey of near-surface seawater

isotopic variability over an extensive geographic area. All seawater collection bottles were conditioned by filling and emptying the bottles three times prior to water sampling. Samples were then stored in 30 mL glass vials with PolyCone caps and sealed with parafilm. Samples were stored at ambient temperature prior to analysis at the University of New Mexico Center for Stable Isotopes.

**Atalante**

The Atalante seawater sampling strategy targeted both near-surface isotopic variability and isotopic variability with depth. A total of 114 samples of seawater were collected for isotopic analysis over 27 days between 23 January and 18 February (DOY 23-49). Sixty-three near-surface samples were collected (daily or sub-daily) from a faucet associated with the thermosalinograph measuring the ship's water intake at a depth of 5 m. An additional 51 samples were collected from CTD casts at varying depths. The samples were stored in 30 mL amber-glass vials with special fitted caps. The caps were not secured otherwise with parafilm. They were also exposed to high temperatures in transit back to the laboratory – the possible effects of which are discussed in Sect. 3. The samples, which were analyzed at the LCISE facility of OSU Ecce Terra in France, are part of a multidecadal analysis of water isotope research cruise data (waterisotopes-CISE-LOCEAN, 2021; Reverdin et al., 2022).

**3 Data post-processing and uncertainties**

In this section, we provide a detailed report of any corrections, adjustments, masks, or flags applied to the EUREC[4]A-iso data and describe key uncertainties that may affect their quality or interpretation. We also describe any anomalous data points or sampling periods. All datasets provide estimates of both the oxygen and hydrogen isotope ratios of water normalized to the VSMOW-SLAP (Vienna Standard Mean Ocean Water - Standard Light Antarctic Precipitation; Craig, 1961; IAEA, 2017) scale and expressed in units permil. For some platforms, estimates of $d$ – or the deuterium excess parameter, defined as $d = \delta D - 8 \times \delta^{18}O$ (Dansgaard, 1964) – and its uncertainty are provided as well. The water vapor datasets additionally contain estimates of the water vapor concentration, which, unless otherwise specified, is given in the original measurement format as a mole fraction relative to total (moist) air in ppmv (i.e. $n_{wv}/n_t$). Several datasets provide more than one expression of the water vapor concentration (e.g. mole fraction and specific humidity) for ease of comparison with other humidity sensors on the same platform, dropsondes, and radiosondes.

**3.1 Post-processing and uncertainties for water vapor measurements**

While the isotopic analyzers deployed during EUREC[4]A were designed to measure over a large concentration range (e.g. 1,000-50,000 ppmv for the Picarro L2130-i; 5,000-30,000 ppmv for the Picarro L1115-i), lab-based calibrations and field-based comparisons suggest that their humidity measurements are stable and reliable even in more arid conditions, such as typify the free troposphere (Pincus et al., 2021; Sodemann et al., 2017). Pincus et al. (2021), for example, reported errors in 1 Hz water vapor concentration data of <1% for concentrations spanning 200 to 30,000 ppmv. Previous ground-based deployments have also demonstrated that the humidity measurements from these analyzers can be stable over years (Bailey et al., 2015). More detailed discussions on the quality of the water vapor concentrations from the airborne isotopic analyzers can be found in other EUREC[4]A special issue data papers (Pincus et al., 2021; Bony et al., 2022), with only key information repeated here. For all platforms, water vapor concentrations were evaluated in one of two customary ways: through lab-based calibration checks and/or through comparisons with other humidity sensors on the same platform. Any corrections applied to the water vapor concentration data are discussed below.

In comparison, because of the diverse environmental conditions encountered and the various types (and versions) of laser spectrometers used, post-processing of the water vapor isotopic measurements varied widely during

EUREC[4]A. For each platform, we discuss how in-field and/or pre- and post-campaign calibrations were used to normalize the data to the VSMOW-SLAP scale. We also describe how instrumental drift was evaluated and whether it required adjustments to the VSMOW-SLAP normalization over the course of the campaign. Moreover, we discuss any corrections made for known biases associated with low water vapor concentrations (Aemisegger et al., 2012; Bailey et al., 2015) and report any adjustments to timestamps to account for time delays in the measurement systems. The post-processing corrections ensure that measurements from any one platform are not only self-consistent but also comparable with measurements from other EUREC[4]A platforms or with isotopic data from previous field deployments. Uncertainty estimates have been developed that account for either the errors associated with calibrations or the variability in the trade-wind environment. Accounting for these uncertainties and excluding data flagged for quality concerns will ensure that cross-platform comparisons are as accurate as possible. Datasets from the airborne sensors include isotopic information at 1 s resolution, while datasets from the BCO and ships provide isotopic information at 1 or 2 minute resolution.

**BCO (ground-based)**

Water vapor isotope data from both the CRDS and OA-ICOS systems at the BCO were normalized to the VSMOW-SLAP scale following the IAEA's procedure (IAEA, 2017). The two most enriched standards introduced during the in-field calibration checks were used for this purpose (Supplemental Information). Out of the total 20-to-60 minutes a standard was measured, only the most stable part (e.g. 10-to-30 minutes in length) was selected for normalization. The average precision of these stable periods was 0.2‰ for $\delta^{18}O$ and 0.9‰ for $\delta D$.

Variations in the measured standard values from one day to the next were used to assess instrumental drift. For the CRDS system, the drift was estimated at $(0.2\pm0.1)$‰ day$^{-1}$ for $\delta^{18}O$ and $(2.1\pm2)$‰ day$^{-1}$ for $\delta D$ – on par with or just slightly larger than the average precision of the calibration measurements. A linear interpolation between daily calibration checks was used to correct both instruments for the small drift detected.

Because previous studies had shown that the accuracy of the CRDS analyzer's isotopic measurements are independent of water vapor concentration in the humidity range typical of Barbados' tropical environment (e.g. 20,000-28,000 ppmv; Aemisegger et al., 2012), no humidity-dependence correction was applied to either the CRDS or OA-ICOS data. A post-campaign laboratory test conducted using a bubbler system verified the validity of this choice for the CRDS system (Supplemental Information). The humidity dependence of the OA-ICOS isotopic measurements during EUREC[4]A was not explicitly characterized due to the breakdown of the WVISS calibration system; however, subsequent comparisons with the CRDS system (Sect. 4; Fig. S3) and analyses of data collected during previous deployments (Galewsky, 2021) suggest that a significant humidity dependence may have influenced its isotopic measurements. The effects of this water vapor concentration dependence are discussed more thoroughly in Sect. 4.

Based on error propagation from the normalization to VSMOW-SLAP and the drift correction, isotopic measurement uncertainties for the BCO CRDS data at 1 minute time resolution are 1.0‰, 3.0‰ and 3.1‰ for $\delta^{18}O$, $\delta D$, and $d$, respectively. Equivalent uncertainty estimates for the OA-ICOS system are 0.41‰, 0.94‰, and 3.31‰; however, these values may underestimate the total measurement uncertainty, given the possible remnant of a humidity-dependent bias in the OA-ICOS isotopic data. Data users desiring to be more cautious can add the average isotopic differences between the two BCO analyzers to the given OA-ICOS uncertainties (see Sect. 4; Phillips et al., 1997) to derive a more conservative total uncertainty estimate. To gap-fill the BCO time series, data users may consider scaling the OA-ICOS isotope ratios to the CRDS isotope ratios, for example, by applying a mean offset or by means of a regression model (Fig. S4-S5).

As noted in the OA-ICOS dataset's accompanying README file, data users should also be aware that the OA-ICOS $\delta^{18}O$ exhibited large oscillations – on the order of 1‰ – during some periods of ambient sampling. While the

cause of these oscillations has yet to be identified, the fact that they appear only in one isotope ratio and not the other indicates an intermittent problem with the internal spectroscopy. The effect of this oscillation is not included in the OA-ICOS uncertainty estimates, since these estimates are designed to represent trustworthy sampling periods. Data users should be aware that the oscillations in OA-ICOS $\delta^{18}O$ also influence the OA-ICOS $d$ time series.

Water vapor concentrations were corrected for the CRDS system but not the OA-ICOS system. For the CRDS analyzer, an independent linear scaling was applied (Supplemental Information) to adjust for a high bias in the range 10,000 to 30,000 ppmv – which was determined after the campaign using a dew point generator – and to simultaneously convert wet mole fractions ($n_{wv}/n_t$) to dry mole fractions ($n_{wv}/[n_t-n_{wv}]$). Given the known small drift of the CRDS system's humidity measurements (<50 ppmv per month), the bias was assumed constant over the
course of the EUREC$^4$A deployment. The uncertainty of the corrected CRDS humidity measurements is 223 ppmv (dry mole fraction).

Masked (missing) data in either BCO water vapor isotopic dataset represent periods when daily calibration checks or instrument maintenance were performed. All variables have been averaged in 1 minute intervals.

**ATR (airborne)**

The post-processing for the ATR 1 s water vapor isotope data closely follows the procedure presented and applied in previous experiments using the same instrument (Aemisegger et al., 2012; Sodemann et al., 2017; Thurnherr et al., 2020). Normalization to VSMOW-SLAP was performed by applying the scale and offset derived from fitting a linear regression between the measured and known values of the three standards used during the in-field calibration
checks. Standard measurements were deemed of sufficient quality to include in the regression if at least 3 minutes within the calibration period presented no significant drift and exhibited standard deviations less than 1‰ in $\delta^{18}O$, less than 2‰ in $\delta D$, and less than 3000 ppmv in water vapor concentration. Importantly, the in-flight calibration checks did not show a significant difference in either mean or standard deviation compared to the ground-based calibration checks. This lent confidence to the decision to include the ground-based calibration measurements when
evaluating biases and uncertainty in the airborne data. Thirty-two total calibration points were included in the linear fit.

Because the measured drift between flight days (±0.5‰ day$^{-1}$ for $\delta^{18}O$ and ±1‰ day$^{-1}$ for $\delta D$) was of comparable amplitude or smaller than the calibration measurement uncertainty (0.3‰ for $\delta^{18}O$ and 1.25‰ for $\delta D$), no drift
correction was made to the ATR isotopic data. However, three additional corrections were applied based on post-campaign analyses and calibrations performed in August 2020 and March 2021:

1.  Prior to normalization, isotopic biases associated with low water vapor concentrations (<10,000 ppmv) were eliminated by applying a two dimensional fit that accounts for both the water vapor concentration and
565         its isotopic composition (Fig. 7a; cf. Weng et al., 2020). At high flow rates and isotopic values exceeding – 30‰ in $\delta^{18}O$ and –260‰ in $\delta D$, these biases were found to depend only on the water concentration and not on the isotope ratio (Thurnherr et al., 2020). The biases were quantified using three liquid standards, which were converted to gas phase and delivered to the CRDS analyzer in distinct concentrations using a custom-built bubbler system (Supplemental Information; Fig. S6).
2.  The analyzer's water vapor concentrations were corrected and converted from wet ($n_{wv}/n_t$) to dry mole fractions ($n_{wv}/[n_t-n_{wv}]$) by applying a linear scaling determined using a dew point generator.

3.  The isotopic and water vapor concentration time series were shifted to account for time delays and to align
575         the CRDS data with other ATR measurements. A time shift of 15 s – which was determined by lag-correlating the humidity measurements from the isotopic analyzer with those from the plane's dew point

hygrometer – was applied to both isotope ratios and the water vapor concentration. The δD time series was further shifted by an additional 5 s to account for the higher adsorption tendency of the HDO molecule on tubing surfaces, which causes a slower time response (Aemisegger et al., 2012). Shifting the δD time series in this manner produced a correlation of 0.995 with $\delta^{18}$O.

Further details about the ATR calibration measurements and corrections, as well as a schematic of the custom-built bubbler system used for evaluating the isotopic humidity dependence, are provided in the Supplemental Information.

Total uncertainties in the ATR isotopic measurements are 0.8‰, 1.7‰, and 1.9‰ for $\delta^{18}$O, δD, and $d$, respectively. These estimates are valid for water vapor concentrations of 25,000 ppmv, which represent near-surface conditions near Barbados. Isotopic uncertainties increase as water vapor concentrations decrease but do not appear to depend on the isotopic composition of the vapor (Fig. 7b).

Measurements of suspect quality, including those influenced by inlet wetting, are noted in the YAML files that accompany the dataset. See Bony et al. (2022) for a general description of these.

**P-3 (airborne)**

Water vapor isotope ratios from the P-3 were normalized to the VSMOW-SLAP scale by fitting a linear regression between the measured and known values of four liquid water standards, which were manually injected into a Picarro vaporizer during a single post-campaign calibration. Each standard was injected at least 9 times, and the first 5-to-6 injections were excluded from the regression fit (Supplemental Information). Higher-than-expected uncertainties in the in-field calibration checks discouraged the use of these measurements for calibration purposes and also precluded the detection of any instrumental drift. Furthermore, although the P-3 water vapor isotope data were tested for dependencies on water vapor concentration both before and after the campaign, no consistent bias could be detected at low water vapor concentrations. Therefore, no humidity-dependence correction was applied. Corrected data are available both at sample rate (~5 Hz) and averaged to 1 s to align with other P-3 airborne datasets (Pincus et al. 2021).

Uncertainty estimates for the P-3 isotopic data follow guidance from the US National Institute for Standards and Technology (NIST, 2021) and account for all of the following: errors in the correction function used to normalize the data, ambigüities about the stability of the normalization with time (i.e. drift), and uncertainties associated with an inconsistent humidity-dependent bias. They represent the addition in quadrature of the residual standard deviations from the pre- and post-campaign humidity-dependence tests and the isotopic uncertainties associated with the normalization function and its stability, with the latter given by

$$\delta_{u.normalization} = \frac{1}{\sqrt{3}} \times max\_difference \qquad (3)$$

where $max\_difference$ represents the maximal difference between normalization correction functions derived in the field and the correction function derived after the campaign's conclusion.

As shown in Fig. 8, calibration uncertainty estimates for the sample-rate P-3 isotopic data (black lines) increase as water vapor concentration decreases. The standard deviations of the 1 s data (blue lines) are of comparable magnitude. Because the calibration uncertainty estimates in the 1 s datafiles are reduced through averaging (by a factor of $1/\sqrt{n}$, where $n$ are the number of sample-rate points per 1 second average), data users wishing to be extra conservative may consider using the 1 s standard deviations for their measure of uncertainty instead.

Even with such extra precautions, actual uncertainties at low water vapor concentrations are likely underestimated for at least three reasons. First, the P-3 analyzer's normalization cannot be verified for isotopic values lower than the most depleted standard used (e.g. $\delta D < -355‰$). Second, despite finding no consistent isotopic humidity dependence in laboratory tests conducted before and after the campaign, there is an obvious shifting bias in $\delta^{18}O$ over the course of the field deployment (which affects calculations of $d$). Figure 9 shows the effect of this transitory bias for research flight 8, where positive $\delta^{18}O$ values in the free troposphere are clearly unphysical. Finally, adsorption and mixing of water vapor within the aircraft sample line reduces isotopic accuracy by slowing the time response and weakening the signal of the isotopic measurements. These effects are much greater for $\delta D$ compared to $\delta^{18}O$ and are particularly evident in the low humidity conditions found at higher altitudes (Fig. 9).

Based on both quantifiable (Fig. 8) and unquantifiable (Fig. 9) measurement uncertainties, we recommend that applications requiring a single isotope ratio use $\delta D$ from the P-3; however, care should be taken at altitudes where water vapor concentrations and isotope ratios are especially low (e.g. > 5000 m). Time periods when $\delta^{18}O$ is clearly suspect have been marked with a quality-control flag in the 1 s data. Periods when both isotope ratios are masked (missing) reflect periods when the analyzer sampled from the dew point generator or when the aircraft was taking off.

No time adjustment has been applied to the P-3 isotopic data to account for delays associated with the flow rate through the sample line. Instead, users are encouraged to apply the time correction described in Pincus et al. (2021) if desirable for their application. The correction for water vapor concentration is also described in Pincus et al. (2021), with additional details provided in the Supplemental Information.

**Meteor (ship-based)**

Water vapor isotope ratios from the Meteor were normalized to the VSMOW-SLAP scale and corrected for drift by linearly interpolating the measurements from the daily calibration checks to the observational sampling rate of 1 Hz. Each ambient data point was then corrected using a unique linear model derived by fitting the interpolated measurements to the known values of the four liquid standards used.

Prior to normalization, the isotopic observations were also corrected for small humidity-dependent biases of up to 0.24 ‰ in $\delta^{18}O$ and 0.36 ‰ in $\delta D$. These biases were determined shortly after the analyzer was installed on the ship using the SDM to generate water vapor from two liquid standards across a range of eight concentrations spanning 19,500 - 35,000 ppmv. No correction was applied to the water vapor concentration measurements since the campaign-mean specific humidity values from the isotopic analyzer and the ship's main meteorological station differed by only 0.13 g kg$^{-1}$.

Total uncertainties in the Meteor's isotopic measurements were estimated by summing in quadrature the bulk uncertainties associated with the liquid standards used to generate reference gas ($\delta^{18}O$, $\delta D$ = 0.14 ‰, 0.69‰), the standard deviations of the residuals from the humidity-dependence correction ($\delta^{18}O$, $\delta D$ = 0.10 ‰, 0.32 ‰), the average precision of the individual calibration measurement periods ($\delta^{18}O$, $\delta D$ = 0.14 ‰, 0.83 ‰), and the variability in the mean measured calibration values over the course of the campaign ($\delta^{18}O$, $\delta D$ = 0.18 ‰, 0.50 ‰). Average values for the uncertainty estimates are 0.29 ‰ for $\delta^{18}O$ and 1.24‰ for $\delta D$.

All variables included in the Meteor water vapor isotopic data files have been averaged to 1 minute and have been masked (removed) for periods when daily calibration checks or instrument maintenance were performed. Data users should also be aware that rain events coincided with some of the largest variations in vapor isotopic composition and with some of the highest values of $\delta^{18}O$ and $\delta D$ observed by the Meteor (Fig. S7). It is not known if these values represent environmental processes, such as the sampling of cold pool air masses, or reflect evaporation of rainwater from the ship's surfaces; hence, caution is advised when analyzing these periods. To assist with data interpretation, a

quality control flag is included marking periods with measured rainfall, the three hours following rainfall, and periods when the risk of contamination from ship emissions increased due to the wind's direction relative to the ship's orientation.

**Brown (ship-based)**

Water vapor isotopic data from the Brown, like those for the P-3, were corrected by applying a single linear function to normalize the data to the VSMOW-SLAP scale. The normalization correction function was derived by regressing
the measured and known values of the three liquid standards from the single in-field calibration check. (Only the last three injections of each standard were included in the regression fit.) Although instrumental drift could not be evaluated during the research cruise, longer subsequent experiments suggest the normalization should have been stable for the 16 day measurement period aboard the Brown. Moreover, based on comparisons with the P-3 analyzer, and given the high humidity range (17,500 pppmv - 28,300 ppmv) in which the Brown sampled, biases associated
with water vapor concentration were assumed negligible, and no humidity-dependence correction was applied. The normalized data are available both at sample rate (~5 Hz) and in 1 minute averages, which align with other Brown meteorological data (Quinn et al., 2021).

Although only a single calibration was performed for the analyzer aboard the Brown, discrepancies in replicate
laboratory measurements of the tertiary standards used during the calibration allow for a fairly large range of plausible linear normalization functions and thus large uncertainty estimates for the Brown isotopic data. The coefficients for the normalization function used to correct the data are based on the average laboratory results, with values of $\{\beta_0 = 1.26‰, \beta_1 = 0.98‰\}$ for $\delta^{18}O$ and $\{\beta_0 = 5.89‰, \beta_1 = 0.97‰\}$ for $\delta D$. However, coefficients as different as $\{\beta_0 = 0.87‰, \beta_1 = 0.96‰\}$ and $\{\beta_0 = 1.85‰, \beta_1 = 1.01‰\}$ for $\delta^{18}O$ and $\{\beta_0 = 4.96‰, \beta_1 = 0.96‰\}$ and
$\{\beta_0 = 6.60‰, \beta_1 = 0.97‰\}$ for $\delta D$ are also justifiable. Uncertainties in the Brown normalization are thus estimated using Equation 3 by considering the maximal difference between plausible normalization curves for the range of isotope ratios measured in the marine environment near Barbados.

For the sample-rate measurements, the estimated normalization uncertainties are 1.15 ‰ for $\delta^{18}O$ and 0.89‰ for $\delta D$.
These uncertainties are reduced when the data are averaged to 1 minute but by less than expected for an instrument with nominal 5 Hz sampling frequency. Because of strong lag 1 autocorrelation in the time series of both isotope ratios ($r$=0.80 for $\delta^{18}O$, 0.83 for $\delta D$), the effective degrees of freedom are closer to 34 and 29 (rather than 308), which results in calibration-related uncertainty estimates of 0.20 ‰ and 0.17 ‰ for 1 minute averages of $\delta^{18}O$ and $\delta D$, respectively. The standard deviations associated with the 1 minute averages are typically higher and thus may be
a preferred estimate of measurement uncertainty, likely because they also reflect variability in the environment.

Because the water vapor concentrations from the Brown analyzer were not calibrated before deployment, they are reported as measured. However, a comparison in 1 minute intervals with the ship's primary specific humidity measurement (*qair*; Quinn et al., 2021) suggests a median difference of just 0.20 g/kg for all periods when the ship's
contamination flag is 0. This is equivalent to a potential positive bias of 320 ppmv in water vapor mole fraction. Additional comparisons between the isotopic analyzer's humidity measurements and the 10 Hz LI-COR on the ship (C. Fairall and E. Thompson, personal communication, 2020) were used to shift the isotopic analyzer's time series (following the formula 69.94 - $2.51\times10^{-5} t$) to address a drifting offset of 13.8 to -18.5 seconds over the course of the campaign.
The 1 minute isotopic data files contain a contamination flag equivalent to that found in the Brown meteorological dataset, where a non-zero value marks periods of potential contamination (Quinn et al., 2021). Optimal sampling periods occurred when the Brown was pointed into the wind, minimizing contamination by the ship's stack aft of the aerosol container in which the analyzer was housed. A flag value of 2 has been added to the isotopic files to mark
time periods when the Brown was near port, when other meteorological data are not reported. An additional flag has

been added to mark periods when the blower, pulling air through the sampling mast, into the aerosol container, was reversed.

**Atalante (ship-based)**

Atalante water vapor isotopic measurements were normalized to VSMOW-SLAP based on a single post-campaign calibration, by fitting a simple linear regression between the measured and known values of three liquid standards. The effects of instrumental drift were addressed by linearly interpolating the single liquid standard measured daily during the cruise to each ambient observation and subtracting the differences from the known standard value. Three anomalous calibration checks were ignored in this procedure – those made on 26, 29, and 30 January 2020 (DOY 26, 29, 30) – which may have been affected by shifts in the liquid standards themselves. However, since shifts in the instrument's spectroscopy cannot be ruled out, it is possible that the time series for the period 26 to 30 January 2020 (DOY 26-30) could be in error by approximately 0.4‰ in $\delta^{18}O$ and 1.7‰ in $\delta D$.

Prior to normalization, the isotopic data were also corrected for dependencies on water vapor concentration, which were assessed both before and after the cruise by generating reference gas from a single standard across a range of water vapor concentrations. Additional corrections include a uniform 0.6 ‰ offset added to $\delta D$ based on a suspected issue with the standard values and a 2% scaling of the CRDS water vapor concentrations based on a comparison with the ship's main meteorological station. The corrected CRDS variables reported in the Atalante datafiles are all averaged in 2 minute intervals.

Estimates of uncertainty for both the water vapor concentration and its isotopic composition are provided by the standard deviations associated with the 2 minute averages. However, as with other platforms, actual uncertainties may be larger. Because in-field calibration checks relied on a single standard – one that was more depleted than the typical ambient vapor sampled – biases in the Atalante isotope ratios may be underestimated. How representative the water vapor measurements are of the environment also remains to be evaluated thoroughly. Although the analyzer's inlet was positioned away from any vent on the ship, air from the ship's interior could have influenced the isotopic measurements at times. It is also not clear from which altitude air entering the analyzer would have originated and whether this would have depended on the direction of the wind relative to the ship. Nevertheless, satisfactory agreement between the isotopic analyzer's water vapor concentrations with the ship's main meteorological station allays some of these concerns.

Masked (missing) measurements in the Atalante vapor isotopic dataset include times during which the analyzer sampled reference gas and the period 0037 to 2212 UTC on 26 January (DOY 26), during which time the analyzer was not functioning properly. Poor data quality periods have been flagged, as have periods when the water vapor isotopic measurements were likely influenced by precipitation or by exhaust or recycled air from the ship (0.4% of the data).

**3.2 Uncertainties for rain and seawater samples**

Rain and seawater samples were analyzed in established isotopic laboratories following the EUREC[4]A deployment. Rainwater isotope ratios for the BCO, Atalante, and Merian were measured with a Picarro L2130-i at the isotope laboratory at the University of Freiburg. Atalante seawater isotope ratios were analyzed at the LCISE facility of OSU Ecce Terra in France. And, rain and seawater isotope ratios from the Meteor and Brown were measured on a Picarro L2140-i at the University of New Mexico's Center for Stable Isotopes. (Analysis of Brown seawater samples is still ongoing.) Reported uncertainties (Table 1) are thus the analytical uncertainties associated with the long-term accuracy of the liquid standards used to determine the isotope ratios of each sample.

Additional (unquantified) uncertainties may stem from small-scale variability in rainfall intensity and isotopic composition, as well as from post-sampling evaporation. One study of 10 European precipitation events, using an array of samplers similar to the Palmex RS1, found uncertainties related to such factors to be <0.3‰ in $\delta^{18}$O and <2‰ in $\delta$D (Fischer et al., 2019). However, variations in rainwater collection times and sample storage conditions during EUREC$^4$A may have increased the likelihood that post-sampling evaporation occurred, resulting in larger

biases than those reported by Fischer et al. (2019).

On the Brown, for example, several rainwater samples were not collected until up to a few days after precipitation had ended. Other samples may include catch from multiple storms, making it hard to gauge exactly how long rainwater remained in the sampler. For samples where it is known that collection was delayed for more than seven

hours, flags are provided in the data file. On average, these samples have higher isotope ratios than those collected soon after precipitation ended, supporting our suspicion that post-sampling evaporation occurred (see Sect. 4). Liquid samples from the Brown also remained in storage, without temperature regulation, in the ship's aerosol container for over a year due to access and shipping complications associated with the COVID-19 pandemic; however, there is no clear evidence that the full batch of rain and seawater samples experienced evaporative

enrichment while sealed with Polycone caps and parafilm.

Twenty-three of the 114 seawater samples analyzed from the Atalante have been flagged for potential post-sampling evaporation. It is believed that these samples "breathed" through leaky caps during storage in high temperature conditions at the port of Pointe à Pitre, where they remained for two months. The flagged samples were corrected

using an empirical relationship between $d$ and salinity that was based on previous sampling in the trade-wind region. Higher-than-expected $d$ was then used to bias-correct $\delta^{18}$O and $\delta$D, by assuming a 1:2 relationship between the two isotope ratios. Consequently, uncertainties in the isotopic estimates for these samples may be as large as 0.1‰ and 0.15‰ for $\delta^{18}$O and $\delta$D, respectively. Isotope ratios for the first Atalante rainwater sample are also excessively high, indicating that this particular sample may have undergone evaporation prior to analysis in the laboratory.


In contrast, although the BCO and Merian precipitation datasets include flags to mark sample collections that occurred one hour or more after rain had stopped, close examination of the data suggests post-sampling evaporation is not a concern for these samples. The BCO flag can be used instead to discriminate between event-based samples (collection ≥ 1 hour after rain) and those collected every 10 minutes during the trailing cold front on 22 January

(DOY 22). Both the BCO and Merian datasets also include a flag to mark times when samples were collected from the Palmex RS1 sampler, but no precipitation was detected by the platform's primary weather station.

**4 Cross-platform data comparisons and opportunities**

EUREC$^4$A's extensive isotopic measurement network provides ample opportunity to examine spatiotemporal

variability in the trade-wind environment, as well as to assess the isotopic data quality more thoroughly. Here, we compare isotopic measurements across platforms, including between in-situ and satellite-based sensors, in order to further evaluate estimates of measurement uncertainty. We also describe additional opportunities for isotopic data comparisons for future study. Overall, we find strong coherence in EUREC$^4$A isotope ratios when comparing long averages or samples from similar air masses, so long as data of suspect quality are excluded. For an in-depth

comparison of the humidity measurements among platforms, we refer readers to publications that have addressed this aspect in detail (Pincus et al., 2021; Bony et al. 2022). Information about other meteorological information collected during the field campaign period, including data from dropsondes (George et al., 2021) and radiosondes (Stephan et al., 2021), can be found in other EUREC$^4$A special issue papers in *Earth System Science Data*.

**4.1 In-situ isotopic data comparisons**

**4.1.1 Surface water vapor, rain, and seawater**

Campaign-mean isotope ratios in near-surface water vapor, precipitation, and seawater are highly consistent across platforms (Fig. 10, 11), even though data from each platform cover distinct locations and time periods. This consistency is especially impressive for the near-surface water vapor measurements (Fig. 10), which are less effective in integrating regional signals and tend to have higher uncertainties than the liquid water samples. Even

where differences do exist, many of these match theoretical expectations, at least for measurements taken over open water. For example, near-surface water vapor values over the Atlantic exhibit a subtle depletion – most evident in $\delta^{18}O$ – from the southernmost latitudes (the Atalante) north to Tradewind Alley (the Meteor and Brown) and up to the aircraft legs at 150 ±15 m (ATR and P-3). These patterns are consistent with the idea that isotope ratios tend to decrease with latitude and altitude, due to variations in temperature and water vapor concentration (Dansgaard,

1964), lending confidence to the measurement accuracy.

Near-surface values from the land surface are somewhat less consistent than those from the oceanic environment. Most surprising is the fact that the BCO's two analyzers differ by 1.05 and 3.53‰ in terms of campaign-mean $\delta^{18}O$ and $\delta D$, respectively, even though they sampled from the same inlet and were normalized to VSMOW-SLAP and

drift corrected using the same standards and procedure (Fig. 10; cf. Table 2 RMSEs, which provide another measure of difference). While the BCO OA-ICOS values are similar to near-surface water vapor isotope ratios measured at sea and at the Grantley Adams International Airport (approximately 14 km to the southwest of the BCO), we are confident that the higher isotope ratios of the BCO CRDS system are more accurate. We believe the higher values of the CRDS analyzer are the result of sea spray evaporation associated with wave breaking at Barbados' most

windward point, making these data uniquely suited to evaluate the influence of this mechanism on the local air-sea flux.

In contrast, the lower isotope ratios of the BCO's OA-ICOS analyzer likely reflect an uncorrected water vapor concentration bias that can be significant for OA-ICOS systems even at high humidity levels (Sturm and Knohl,

2010; Supplemental Information). Indeed, a recent analysis from the Azores suggests this bias may be about 1‰ in $\delta^{18}O$ and 3‰ in $\delta D$ at humidity levels typical of Barbados (Galewsky, 2021). Both because of this bias and the suspected problems with instrument spectroscopy (Sect. 3.1), we recommend that scientific analyses use the CRDS time series preferentially. The unexpected discrepancy between the BCO analyzers highlights the challenge of accurately characterizing and correcting for all relevant biases in field-deployable water vapor isotopic instruments,

particularly given the distinct sensitivities of different measurement technologies.

Despite their sizable mean offsets, time series from the two BCO analyzers are still strongly correlated for both water vapor concentration and $\delta D$, bolstering our confidence in the variability captured in their respective signals (Fig. 12, Table 2). (Correlation between the $\delta^{18}O$ time series is diminished by the oscillation in the OA-ICOS signal

(see Sect. 3.1).) Low-frequency coherence is also apparent when comparing the time series from the BCO with those measured by nearby ships. The Meteor, for example, was frequently close enough to Barbados' eastern shores that air masses sampled on the ship would have reached the BCO about 9 hours later (assuming easterly wind speeds of about 7 m s$^{-1}$). Shifting the Meteor time series to account for this presumed time difference produces correlations with the CRDS analyzer of 0.4-to-0.5 when 1 hour moving averages are applied to both datasets. These results

suggest that mixed-layer variability in the trade environment is coherent across hundreds of kilometers. Spikes that appear in the Meteor time series but not the land-based datasets represent measurement periods during and after rainfall, which are flagged in the Meteor dataset (previously described in Sect. 3.1; see Fig. S7).

Compared with the near-surface water vapor measurements, campaign-mean rainwater and seawater values are even

more consistent across platforms (Fig. 11), providing compelling evidence that cross-platform differences in sample

storage did not influence the liquid water isotope ratios significantly. The high consistency does, however, depend on taking flags into proper consideration. For example, had Fig. 11 included Brown rainwater samples flagged for late collection times (Sect. 3.2), average rainwater isotope ratios from the Brown would increase from -0.13 to +0.61‰ in $\delta^{18}O$ and from 10.68 to 13.36‰ in $\delta D$, causing the Brown's values to be substantially higher than the other platforms. Delayed collections also explain why the samples taken from aboard the Brown on 24 January (DOY 24) are more than 2.7 and 12.5‰ more enriched in $\delta^{18}O$ and $\delta D$, respectively, than the BCO sample collected 37 km downwind on the same day (Quinn et al., 2021). We suspect Brown samples from 24 January include both fresh precipitation from that day and old precipitation that had undergone evaporation while sitting in the sampler.

In contrast, differences in campaign-mean rainwater values between the BCO and the ships appear to reflect real environmental variability. Indeed, if BCO samples from the trailing cold front (22 January; DOY 22) are excluded from the campaign-mean averages (open symbols; Fig. 11), cross-platform coherence in rainwater values increases. Because rain on 22 January was associated with large-scale convergence, its isotope ratios are much lower than samples representing typical shallow convective showers (cf. Risi et al., 2020), which were likely prevalent where other platforms sampled. Reduced rain evaporation, as a result of more intense precipitation, may have also played a role in lowering BCO precipitation isotope ratios on 22 January and increasing their deuterium excess.

### 4.1.2. Atmospheric vertical profiles

The two airborne isotopic analyzers provide an opportunity to evaluate 3-D isotopic variability in the tropical atmosphere. Ignoring the unphysically high free tropospheric $\delta^{18}O$ from the P-3 (Sect. 3.1), both analyzers show the expected tendency toward isotopic depletion with height and are very similar in value in the lowest well-mixed levels of the boundary layer. In contrast, there are some differences aloft. On average, lower free tropospheric $\delta D$ (approximately 2000-5000 m.a.s.l.) tends to be more depleted on the P-3 compared to the ATR. Arguably, much of this difference is due to the fact that the P-3 experienced a wider range of humidity conditions, having sampled more extensively at higher altitude and across a wider longitudinal range. Supporting this idea is the fact that P-3 takeoffs and landings, which were flown in closest proximity to the HALO circle, are very similar in vertical structure to the ATR, unlike the slantwise ascents and descents and cloud legs flown farther to the east (Fig. 13). The differences between profiles separated in space are similar in magnitude to differences in profiles separated in time (e.g. takeoffs and landings) and thus likely reflect real variations in the humidity structure of the atmosphere.

That said, we still suspect P-3 $\delta D$ may be biased low in the free troposphere. For the earliest research flights, when P-3 $\delta^{18}O$ was more trustworthy at altitude, the lowest $\delta^{18}O$ values observed are consistent with the amount of distillation an air parcel from the local marine boundary layer would have experienced had it ascended pseudoadiabatically. In contrast, the $\delta D$ values are substantially lower than the pseudoadiabatic (i.e. Rayleigh) prediction (not shown). Scientific investigations might thus consider scaling the P-3 $\delta D$ to account for this inconsistency between the analyzer's two isotope ratios.

Estimates of the marine boundary layer isotopic composition – necessary for theoretical predictions of vertical isotopic lapse rates – can be derived not only from the airplanes themselves, but also from the other platforms, either by using the campaign-mean values or observations taken during targeted flyovers. The Meteor and Brown, for example, frequently probed the near-surface oceanic environment over which the ATR and P-3, respectively, flew. Quinn et al. (2021) provide a detailed list of periods during which the Brown was stationed within the P-3 dropsonde circle. Following each circle, the P-3 typically flew a slantwise descent, designed to sample the water vapor isotope ratio profile in the same geographic vicinity. The ATR also conducted targeted flyovers of the BCO and flew near-surface legs, 60 m above the ocean surface, within the HALO circle (Bony et al., 2022).

## 4.2. Remotely sensed and in-situ isotopic data comparisons

While EUREC[4]A's in-situ isotopic measurement network affords numerous opportunities to assess spatial variability in the trade-wind environment, routine satellite retrievals of δD over the study region provide additional large-scale context for the in-situ collections. They also provide compositional information about air masses
upstream of the target measurement region, so long as careful consideration is given to the fact that the remote sensors' sensitivity to the atmosphere differs greatly from that of the in-situ analyzers. Three satellite δD products are available for the EUREC[4]A measurement region and time period as of this writing. NASA's Atmospheric Infrared Sounder (AIRS), aboard the Aqua satellite, provides an estimate of mid-free tropospheric δD, with greatest sensitivity to pressure altitudes between 825-400 hPa (Worden et al., 2019; J. Worden, personal communication,
2020). The European Organisation for the Exploitation of Meteorological Satellites' (EUMETSAT) Infrared Atmospheric Sounding Interferometer (IASI; whose data collectively come from three satellites: Metop-A, Metop-B and Metop-C) provides estimates of mid-tropospheric δD, with greatest sensitivity to altitudes between 2.5 and 6.5 km a.s.l. (Schneider and Hase, 2011; Diekmann et al., 2021c). And, the European Space Agency's (ESA) TROPOspheric Monitoring Instrument (TROPOMI), onboard the Copernicus Sentinel-5 Precursor (S5P) satellite,
provides an estimate of total-column δD (A. Schneider et al., 2022). Total-column isotopic retrievals are dominated by the lowermost altitudes, where most water vapor resides. All three remote sensors also provide retrievals of water vapor concentration.

The two European isotopic products have been repackaged into custom subsets for the EUREC[4]A-iso effort. One
subset provides retrievals within a 10° x 10° box defined by 5°- 15°N and 50°- 60°W. The other covers an extended region to support Lagrangian analyses of air mass transport history (i.e. 21° S - 54° N and 110° W - 22° E; see Villiger et al., 2022). Both subsets cover the period between 10 January and 20 February 2020 (DOY 10-51).

The IASI dataset (generated by the latest version of the MUSICA retrieval algorithm; M. Schneider et al., 2022;
Diekmann et al., 2021c) is customized for the 10°x10° box over Barbados and consists of $H_2O$-δD pairs at all retrieval grid levels between the surface and 56 km, full averaging kernel information, and flag variables indicating the quality of the individual observations. These data are provided with full information for each individual observation (a priori profiles, averaging kernels, uncertainty covariances, etc.). In addition, to reduce data volume and storage needs, $H_2O$-δD pairs for the extended EUREC[4]A-iso region are provided without full averaging kernels
and only for three selected altitudes with high sensitivity: 2.9 km, 4.2 km, and 6.4 km. For both subsets, data are provided over land and ocean but only for cloud-free conditions. Typical uncertainties are 10-30‰ in δD. Data users are referred to M. Schneider et al. (2022) and Diekmann et al. (2021c) for additional information (including the data user guide). The full MUSICA IASI $H_2O$-δD pair dataset can be accessed at https://dx.doi.org/10.35097/415.

TROPOMI's repackaged data contain the following variables for the extended EUREC[4]A-iso region: modified Julian date, longitude, latitude, column-$H_2O$ and -HDO with their retrieval errors, averaging kernels and a priori profiles, a posteriori column-δD and its retrieval error, and a quality flag. The quality flag is 1 for clear-sky scenes, 0.5 for scenes with low clouds (with co-retrieved cloud center height 2 km or less), and 0 for all other scenes. Data with a quality value of 0 should not be used. The median bias – relative to co-located ground-based Fourier
transform infrared (FTIR) observations by the Total Carbon Column Observing Network (TCCON) – is 3 % in $H_2O$ and 17 ‰ in δD for clear-sky scenes and 11 % in $H_2O$ and 20 ‰ in δD for cloudy scenes. A. Schneider et al. (2022) describe the retrieval and provide a validation. The full TROPOMI dataset is available from https://tropomi.grid.surfsara.nl/hdo/.

As demonstrated in Fig. 14, the satellites provide rich spatial context for the in-situ data. Nevertheless, when using the two in tandem, care must be taken to consider differences in what each type of measurement represents. For example, even though TROPOMI's total column estimates are weighted toward the boundary layer, the TROPOMI δD values do not increase toward the equator (Fig. 14a) like the near-surface in-situ values (Fig. 10). Instead, they

vary with the atmosphere's vertical humidity structure, which alters the retrievals' sensitivity to low isotope ratios
aloft. Near Barbados, very depleted free tropospheric δD values have little influence on the total column retrieval
since free tropospheric water vapor concentrations are so low (Fig. 14b). In contrast, in regions where deep
convection regularly moistens the free troposphere, isotope ratios aloft have more influence in lowering the total
column δD.

Other important differences between the remotely sensed and in-situ measurements emerge when comparing vertical
profiles of water vapor and its isotopic composition from IASI, the P-3, and ATR. While IASI detects broad
differences in vertical structure between the trade-wind region and areas equatorward, it misses much of the
finescale variability captured by the airborne sensors (Fig. 15; cf. Stevens et al., 2017). This smoothing is the result
of IASI's wide averaging kernel, which causes measurements at any one pressure altitude to depend strongly on the
atmospheric state at numerous other levels. In contrast, despite resolving greater variability in the vertical, the
aircraft measurements strongly convolve horizontal with vertical information (for instance, because of the way the
P-3 profiled the atmosphere by flying slantwise descents and ascents). Moreover, each aircraft flight provides but a
few distinct snapshots of the atmosphere's isotopic vertical structure, compared to the larger number of satellite
retrievals within a given region.

Direct comparisons between the airborne and space-based measurements should therefore consider carefully how
best to aggregate the data in space and time. For the most accurate comparison, the best practice is to apply the
satellite instrument's averaging kernels to the in-situ profiles (e.g. Schneider et al., 2015). This avoids errors in the
comparison caused by the limited vertical sensitivity of the satellite retrievals. However, when assessing the vertical
structure of the atmosphere from a process-based perspective, averaging kernels can smooth out interesting
structures in the aircraft data, such as elevated moist layers or dry tongues, complicating their interpretation.

## 5 Data availability

All EUREC[4]A in-situ water isotopic data and the repackaged IASI and TROPOMI products are available through
the AERIS portal (https://eurec4a.aeris-data.fr/). Data from the P-3 and Brown are also archived at the National
Centers for Environmental Information (https://www.ncei.noaa.gov/). Individual datasets, which have been created
for each platform and sample type (e.g. water vapor, precipitation, seawater), are listed with their DOIs and quality-
flag notes in Table 3. We encourage data users to cite individual datasets; however, the full collection may be
accessed with one search through https://doi.org/10.25326/418 (Bailey et al., 2022).

## 6 Concluding perspective on dataset uses

The collection of water vapor, rainwater, and seawater isotopic data gathered during EUREC[4]A comprises one of
the most extensive cross-platform water isotopic datasets to date. As a result, analyses using datasets specific to
many of EUREC[4]A's airborne and ship-based platforms – as well as the BCO – will benefit from the extra
observational constraint on water cycle processes in the trade-wind region that water isotopes provide (Fig. 1). For
instance, combining microphysical data, such as raindrop size distributions, with precipitation isotopic
measurements could provide a novel way to independently verify rain evaporation rates (cf. Salamalakis et al., 2016;
Graf et al., 2019; Sarkar et al., 2022). Similarly, comparing water isotopic information with moisture flux estimates
– derived from eddy covariance or budget techniques – could provide complementary time-integrated and
instantaneous perspectives on moisture exchange between the ocean and air. Water vapor isotope ratios could also
constrain mixing processes, such as entrainment into the sub-cloud layer, since mixing between atmospheric layers
produces predictable variations in the isotope ratio as a function of water vapor concentration (Noone et al. 2011;
Noone 2012). And, as pseudo-conserved tracers on larger scales, isotope ratios could provide important context for

interpreting anomalies in other atmospheric constituents (trace gases, aerosols) by helping identify the source regions and moisture transport pathways of distinct air masses.

The fact that the EUREC[4]A dataset includes isotopic information for different moisture reservoirs also creates opportunities to evaluate scientific questions that have long interested water isotope researchers. For example, because the tropical marine boundary layer feeds the global water cycle, several recent studies have asked what controls the isotopic composition of this important near-surface layer (Benetti et al., 2018; Risi et al., 2020). These studies have shown that the near-surface atmosphere is more depleted in isotopically heavy moisture than the often

used "closure" assumption suggests. Devised by Merlivat and Jouzel (1979), the "closure" assumption explains variations in marine boundary layer isotope ratios solely in terms of local thermodynamic conditions and evaporation, neglecting the potential influence of entrainment of dry air from the free troposphere above. However, the relatively high isotope ratios it predicts match neither data collected during previous ocean cruises (Benetti et al., 2014; 2018) nor Large-Eddy Simulations (LES; Risi et al., 2020). Testing of alternative frameworks that do account

for free tropospheric entrainment has been hampered by a lack of co-located oceanic, near-surface water vapor, and lower free tropospheric water vapor isotopic data (cf. Benetti et al., 2018). EUREC[4]A's isotopic measurements of seawater, near-surface water vapor from ships, and atmospheric profiles from aircraft provide a unique opportunity to test such frameworks over highly resolved spatial and temporal scales. Moreover, the improved understanding of water, energy, and mass budgets in the sub-cloud layer afforded by EUREC[4]A's many meteorological and

oceanographic observations will help refine estimates of the equivalent water isotopic budget.

The distribution of isotopic measurement platforms across the EUREC[4]A sampling region also lends itself to Lagrangian analyses aimed at studying variations in convective activity and cloudiness as air masses advect westward with the trade winds. Isotope ratios can provide important additional constraints for such case studies,

helping evaluate thermodynamic and microphysical controls on convective development. Typically, the P-3 and Brown sampled the eastern side of the EUREC[4]A domain, while the ATR and Meteor sampled downwind and to the west. All of these platforms measured upwind of the BCO, potentially creating opportunities to track air masses for multiple hours, if not days, at a time. Such analyses could be especially useful for evaluating numerical simulations at the large-eddy scale.


The EUREC[4]A isotopic dataset could also prove useful for evaluating numerical simulations more broadly, such as has been done recently for the eastern subtropical North Atlantic (Diekmann et al., 2021a; Dahinden et al., 2021). Afterall, few water vapor datasets provide vertically resolved isotopic information. Moreover, the limited number of airborne isotopic measurements that existed prior to EUREC[4]A primarily represent higher latitude regions (e.g.

Ehhalt et al., 2005; He and Smith, 1999; Herman et al., 2014; Dryoff et al., 2015; Sodemann et al., 2017; Salmon et al., 2019). EUREC[4]A greatly extends the current small body of observed isotopic profiles from the tropical lower troposphere (cf. Bailey et al., 2013; Herman et al., 2020). Similarly, the rain and seawater samples collected during EUREC[4]A help extend the spatial coverage of existing archives (e.g. Schotterer et al., 1996; Schmidt et al. 1999), providing critical observational benchmarks for model evaluation.


All told, EUREC[4]A facilitated the joint deployment of a number of innovative and experimental measurements to address outstanding questions related to convection and cloudiness in the shallow convective environment of the western Tropical Atlantic (Stevens et al., 2021). The seven in-situ water vapor isotopic datasets, five precipitation isotopic datasets, and three seawater isotopic datasets described in this paper helped contribute to EUREC[4]A's

visionary approach and are openly available for the community to use in evaluating the processes that regulate the shallow convective hydroclimate state.

**Author contribution**

AB, FA, LV, SAL, GR, and EQM coordinated measurements in the field, processed the data, and drafted the manuscript. SB, JG, DN, PKQ, SS, BS, and EJT designed and directed sampling strategies for the isotopic
measurement network and individual platforms. CA, DBB, TB, DC, SPdS, MD, DH, PM, MR provided critical measurement support and guidance on data quality. TB, CJD, AS, and MS provided the remotely sensed data and their descriptions. All authors helped edit and refine the initial manuscript draft.

**Competing interests**

The authors declare that they have no conflict of interest.

**Acknowledgements**

The extensive measurements of water isotope ratios during EUREC[4]A would not have been possible without the many operational and technical staff who supported each research facility, including (but not limited to) captains, pilots, other ship and aircraft crew, technicians, engineers, project managers and administrators. Specific support for seawater collections was provided by Alex de Klerk, Jérôme Demange, and Kyla Drushka. Several students also
assisted with rainwater collections at the BCO. We thank fellow chief scientists Chris Fairall, Janet Intrieri (leg 2 of Brown cruise), Johannes Karstensen, Stefan Kinne, and Wiebke Mohr for their leadership in planning and directing sampling from the various research vessels and aircraft, as well as Peter Blossey, whose vision and networking efforts helped guide the EUREC[4]A-iso endeavor. We also thank our other scientific colleagues who joined us in the field and shared their expertise and enthusiasm. We are grateful to Eric DeWeaver and Brigitte Bauerle for
facilitating a last-minute, mid-campaign swap of isotopic analyzers on the Brown and to Vincent Douet and Tim Boyer for help with data archiving. We thank the CIMH and Marvin Forde (CIMH) for their help with logistics and customs in Barbados. We are also grateful to Friedhelm Jansen (MPI Hamburg) and Mario Mech (University of Cologne) for their very valuable support: from logistics on the island to help with instrument setups at the BCO. We thank Barbara Herbstritt at the University of Freiburg im Breisgau for measuring the isotope composition of the
BCO, Merian, and Atalante rain samples, the LCISE facility of OSU Ecce Terra in France for processing the Atalante seawater samples, and Nicu-Viorel Atudorei and his team at the University of New Mexico's Center for Stable Isotopes for analyzing the Brown and Meteor liquid samples. We are grateful to Mampi Sarkar, Teresa Campos, Mathieu Casado, an anonymous reviewer, and *ESSD* Senior Chief Editor David Carlson for their valuable comments on early drafts of the manuscript and the pre-print discussion paper.

Many funding programs supported the researchers and facilities involved in EUREC[4]A-iso. F.A. and L.V were funded by Swiss National Science Foundation (SNSF) Grant No. 188731, S.L. and G.J. were supported by NSF ATM 1937583, E.Q.M. and S.P.d.S. were supported by NOAA Climate Variability and Predictability Program Award NA19OAR4310375, D.B.B. was supported by Poland's National Science Centre grant no.UMO-
2018/30/M/ST10/00674, M.D. was funded by NSF grant AGS-1938108, and D.H. and D.N. were supported by NSF Climate and Large-Scale Dynamics Award 1937780. In addition, this material is based upon work supported by the National Center for Atmospheric Research, which is a major facility sponsored by the National Science Foundation under Cooperative Agreement 1852977. For D.C. and P.K.Q. this is PMEL contribution number 5324. This work is also a contribution to the LEFE/IMAGO-GMMC project EUREC[4]A-OA, to the JPI-Climate and JPI Oceans project
EUREC[4]A-OA, and to the TOSCA SMOS-Ocean and EUREC[4]A-OA projects supported by CNES (Centre National d'Études Spatiales). Support for these projects was also obtained from IFREMER, the French Research Fleet, the French Research Infrastructures AERIS and ODATIS, IPSL, the "Chaire Chanel" of the Geosciences Department at ENS and the European Union's Horizon 2020 research and innovation program under grant agreement no. 817578

TRIATLAS. The NOAA Climate Program Office, under the Climate Variability and Predictability Program,
supported ATOMIC and the deployment of the P-3 and Brown. European Research Council (ERC) grant No 694768
(EUREC[4]A) funded the ATR operations.

We also thank those programs that supported the satellite data processing. TROPOMI data processing was carried
out on the Dutch national E-infrastructure with the support of the SURF Cooperative. The MUSICA IASI retrievals
were initiated during the project MUSICA (funded by the European Research Council under the European
Community's Seventh Framework Programme (FP7/2007-2013)/ERC Grant Agreement number 256961). The
MUSICA IASI work received financial support through MOTIV and TEDDY (funded by the Deutsche
Forschungsgemeinschaft under project IDs/Geschäftszeichen 290612604/GZ:SCHN1126/2-1 and
416767181/GZ:SCHN1126/5-1, respectively), and INMENSE (funded by the Ministerio de Economía y
Competividad from Spain, CGL2016-80688-P). MUSICA IASI retrieval calculations were performed on the
supercomputers ForHLR and HoreKa, funded by the Ministry of Science, Research and the Arts Baden-
Württemberg and by the German Federal Ministry of Education and Research. The MUSICA IASI work was also
carried out with support from the Teide High-Performance Computing facilities, provided by the Instituto
Tecnológico y de Energías Renovables (ITER), S.A (teidehpc.iter.es).

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

**Tables**

**Table 1.** Bulk uncertainty estimates (in units ‰) for precipitation and seawater isotope ratios.

| Platform | $\delta^{18}O$ | $\delta D$ |
|---|---|---|
| **BCO** | | |
| precipitation | 0.16 | 0.60 |
| **Meteor** | | |
| precipitation | 0.20 | 0.51 |
| seawater | 0.23 | 0.56 |
| **Brown** | | |
| precipitation | 0.20 | 0.80 |
| seawater | -- | -- |
| **Atalante** | | |
| precipitation | 0.16 | 0.60 |
| seawater | $\leqq0.1$ | $\leqq0.15$ |
| **Merian** | | |
| precipitation | 0.16 | 0.60 |


**Table 2.** Root mean square error (RMSE) and Pearson correlation coefficients (CORR) for comparisons between the BCO CRDS time series and the time series of three other analyzers, smoothed with 1 hour moving averages. Meteor data were shifted by 9 h and Atalante data were shifted by 10 minutes to account for the geographic distance between measurement sites.

| Analyzer | Period [UTC] | Statistic | specific humidity [g/kg] | $\delta^{18}O$ [‰] | $\delta D$ [‰] | $d$ [‰] |
|---|---|---|---|---|---|---|
| BCO OA-ICOS | 20 Jan to 15 Feb | RMSE | 0.51 | 1.11 | 3.62 | 5.66 |
| | | CORR | 1.00 | 0.66 | 0.94 | 0.42 |
| Meteor | 20 Jan to 15 Feb | RMSE | 1.12 | 1.05 | 4.31 | 4.48 |
| | | CORR | 0.48 | 0.47 | 0.41 | 0.50 |
| Atalante | 17 Feb 0000 - 0500 | RMSE | 0.88 | 0.97 | 1.79 | 5.36 |
| | | CORR | 0.91 | 0.70 | 0.85 | 0.45 |


**Table 3.** EUREC$^4$A-iso in-situ and remotely sensed datasets.


| Water vapor | | | |
|---|---|---|---|
| Dataset | Link | Citation | Notes |
| BCO CRDS | https://doi.org/10.25326/245 | Villiger et al. (2021a) | 1 minute resolution, $H_2O$ provided as a dry mole fraction and specific humidity |
| BCO OA-ICOS | https://doi.org/10.25326/309 | Galewsky and Los (2020a) | 1 minute resolution, $H_2O$ provided as a wet mole fraction and specific humidity, uncorrected water vapor concentration dependencies likely bias the isotopic data low, $\delta^{18}O$ signal also compromised by spectroscopic oscillation (see README) |
| ATR | https://doi.org/10.25326/244 | Aemisegger et al. (2021b) | 1 second resolution, $H_2O$ provided as a dry mole fraction and specific humidity, YAML files flag poor quality data |
| P-3 | https://doi.org/10.25921/c5yx-7w29 | Bailey et al. (2020) | 1 second resolution, $H_2O$ archived in separate files as a wet mole fraction and (dry) mass mixing ratio, uncertainties for both isotope ratios tend to increase with altitude, quality-control flag provided for $\delta^{18}O$, corrected sample-rate data also available |
| Meteor | https://doi.org/10.25326/83 | Galewsky and Los (2020b) | 1 minute resolution, $H_2O$ provided as a wet mole fraction and specific humidity, isotopic spikes observed following precipitation, flag for precipitation and potential contamination periods provided |
| Brown | https://doi.org/10.25921/s76r-1n85 | Bailey and Noone (2021) | 1 minute resolution, $H_2O$ provided as a wet mole fraction, specific humidity, and (dry) mass mixing ratio, flags for contamination and inlet reversal periods provided, corrected sample-rate data also available |
| Atalante | https://doi.org/10.25326/304 | Reverdin et al. (2021) | 2 minute resolution, $H_2O$ provided as a wet mole fraction and specific humidity, temperature and salinity at 5 m depth included, flags for quality control and precipitation periods provided |
| IASI | https://doi.org/10.25326/262 | Diekmann et al. (2021b) | Full vertical profiles provided for the 10° x 10° box defined by 5°-15° N and 50°- 60° W, select levels provided for the extended region 21° S - 54° N and 110° W - 22° E, $H_2O$ provided as a dry mole fraction, quality flags included, full dataset accessible from https://dx.doi.org/10.35097/415 |
| TROPOMI | https://doi.org/10.25326/306 | Schneider and Borsdorff (2021) | Total column information and quality flags provided for the region 21° S - 54° N and 110° W - 22° E, full dataset accessible from https://tropomi.grid.surfsara.nl/hdo/ |

| Rainwater | | | |
|---|---|---|---|
| Dataset | Link | Citation | Notes |

| BCO | https://doi.org/10.25326/242 | Villiger et al. (2021b) | Event-based except for intensive sampling of a front, rainfall amount and quality-control flags included |
|---|---|---|---|
| Meteor | https://doi.org/10.25326/308 | Galewsky and Los (2020c) | Event-based |
| Brown | https://doi.org/10.25921/bbje-6y41 | Quiñones Meléndez et al. (2022) | Event-based, possible concerns include evaporative enrichment due to delayed collections from the sampler and sea spray contamination, flag for substantially delayed collection times included |
| Atalante | https://doi.org/10.25326/305 | Villger et al. (2021c) | Event-based, rainfall collection times are not exact, sample #1 appears unphysical |
| Merian | https://doi.org/10.25326/243 | Villiger et al. (2021d) | Event-based, quality-control flags included, geographic location missing for one sample |

| Seawater | | | |
|---|---|---|---|
| Dataset | Link | Citation | Notes |
| Meteor | https://doi.org/10.25326/307 | Galewsky and Los (2020d) | Near daily at 10 m depth, except for intensive sampling of a diel period |
| Brown | Preliminary data provided in Supplemental Information | -- | Sub-daily at variable depths, laboratory analysis of samples is still in progress as of this writing |
| Atalante | https://doi.org/10.17882/71186 | waterisotopes-CISE-LOCEAN (2021) | Sub-daily at variable depths, temperature, salinity, and quality-control flags included |

**Figures**

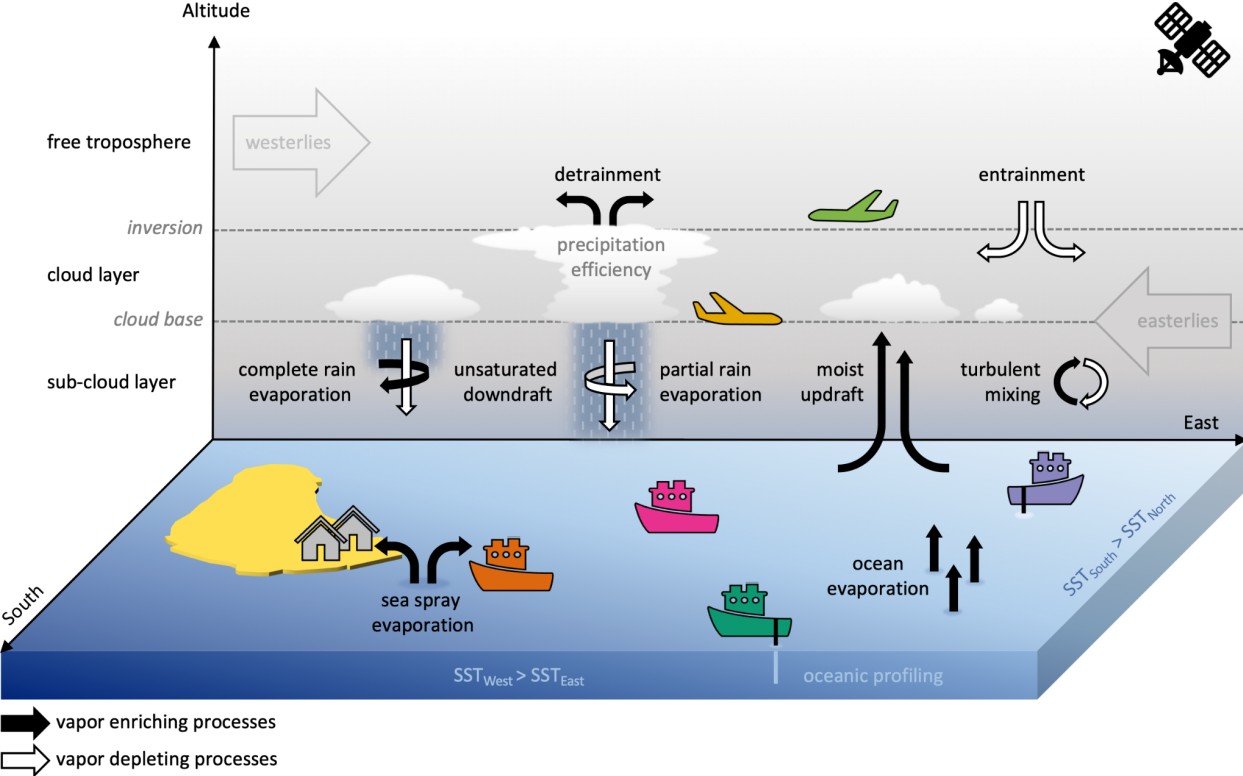

**Figure 1: Examples of moist processes that influence the water vapor isotopic composition of the trade-wind environment around Barbados (yellow landmass) and that can be studied using the EUREC⁴A-iso dataset collection. Black arrows indicate processes expected to increase water vapor isotope ratios, while white arrows indicate processes expected to lower them. The ships and aircraft in the schematic represent the six mobile EUREC⁴A-iso in-situ sampling platforms.**

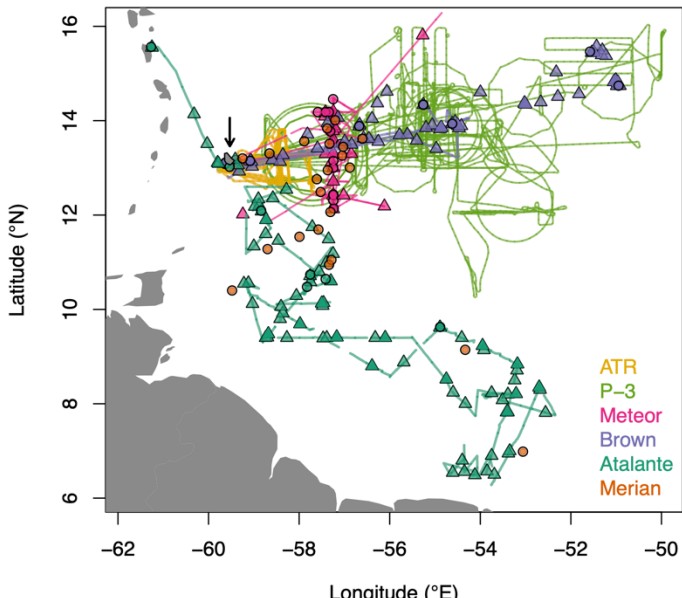

**Figure 2: Map of continuous (water vapor) and discrete (rain and seawater) isotopic sampling during the EUREC⁴A 2020 field experiment. Tracks for the various aircraft and ships are plotted only for periods during which water vapor isotopic sampling occurred. Circles and triangles indicate locations of rain and seawater sampling, respectively. The arrow points to Barbados, whose landmass is outlined in black. Readers are referred to Stevens et al. (2016) for a detailed map of Barbados and a photograph of Deebles Point and the Barbados Cloud Observatory (BCO).**

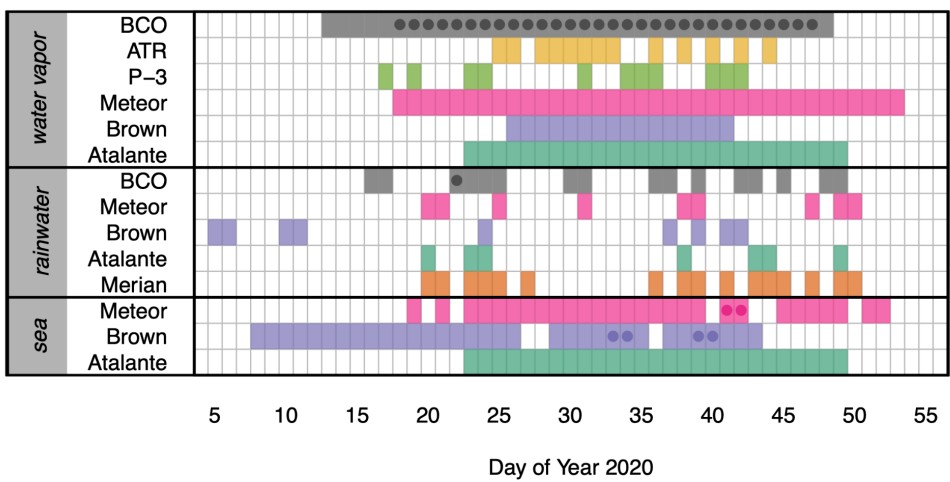

**Figure 3: Timeline of continuous (water vapor) and discrete (rain and seawater) isotopic sampling during EUREC⁴A. Dots either represent days when two laser analyzers were operating at the BCO or indicate intensive observation periods for rain or seawater (see main text for additional details). Discrete samples are represented by their collection times, which, in the case of Brown rainwater, were delayed in some cases by up to several days following precipitation.**

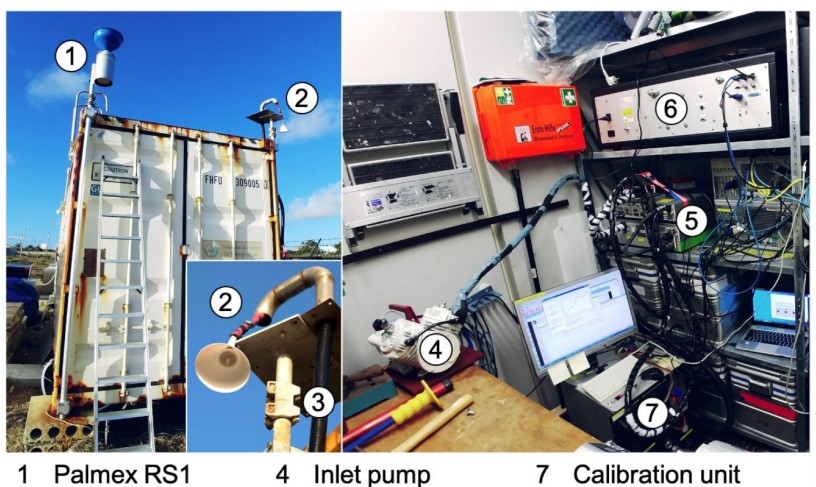

| 1 | Palmex RS1 | 4 | Inlet pump | 7 | Calibration unit |
|---|---|---|---|---|---|
| 2 | Inlet | 5 | Picarro L1115-*i* | | |
| 3 | Heated line | 6 | Los Gatos | | |

1510 **Figure 4: Isotopic sampling installations at the Barbados Cloud Observatory were comprised of a Palmex RS1 rainwater collector (#1) and two water vapor isotopic analyzers (fabricated by Picarro (#5) and Los Gatos Research (#6)).**

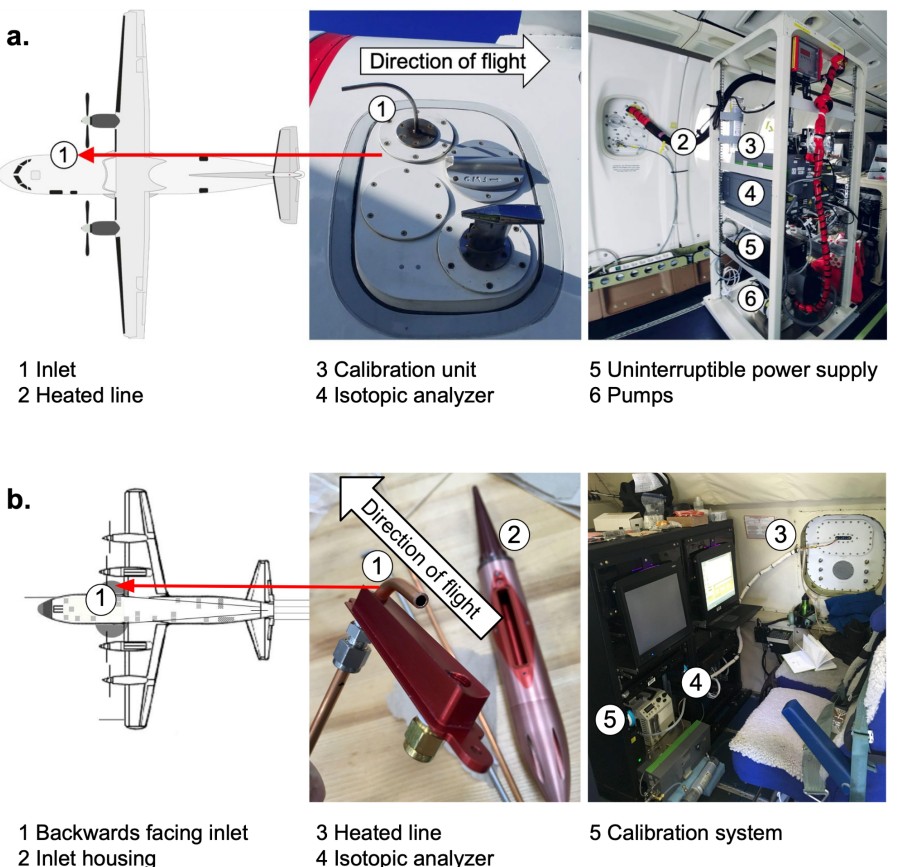

**a.**

1 Inlet
2 Heated line

3 Calibration unit
4 Isotopic analyzer

5 Uninterruptible power supply
6 Pumps

**b.**

1 Backwards facing inlet
2 Inlet housing

3 Heated line
4 Isotopic analyzer

5 Calibration system


**Figure 5: Water vapor isotopic sampling installations on the a) ATR and b) P-3 aircraft. (ATR schematic downloaded from https://t3projects.mpimet.mpg.de/coordination/platform-schematics. P3 schematic provided by NOAA.)**

**Figure 6: Water isotopic sampling installations aboard a) the Meteor, b) the Brown, c) the Atalante, and d) the Merian. (Meteor and Merian ship schematics provided by University of Hamburg. Brown ship schematic provided by NOAA. Atalante ship schematic copyright ©Ifremer. Atalante photos courtesy of Jérôme Demange.)**



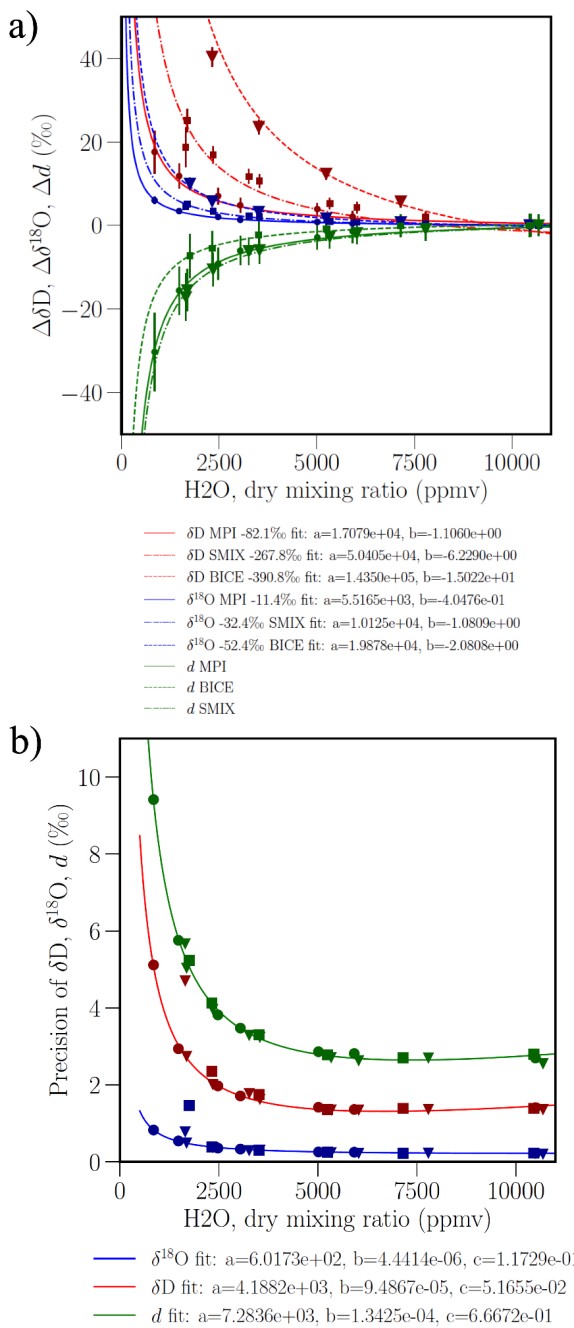

**Figure 7: Bias corrections and uncertainties associated with the ATR water vapor isotopic measurements: a) symbols illustrate the humidity dependence of the isotopic measurements for three distinct liquid standards (MPI, SMIX, BICE) while lines show the correction functions used to remove the detected biases in δ¹⁸O (blue) and δD (red); b) precision of the isotopic measurements as a function of the measured water vapor concentration.**

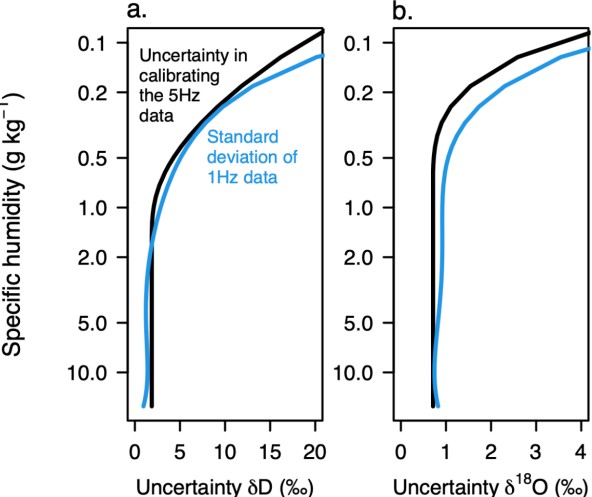

**Figure 8: P-3 isotopic uncertainties (x-axis) plotted as a function of specific humidity (y-axis). The y-axis is plotted on a logarithmic scale to convey the tendency toward higher uncertainty with aircraft altitude. Black lines represent the uncertainties associated with calibrating the 5 Hz data. Blue lines represent the standard deviations associated with the 1 s averages from the first two research flights: they reflect both the variability of the environment and the imprecision of the isotopic analyzer in flight.**


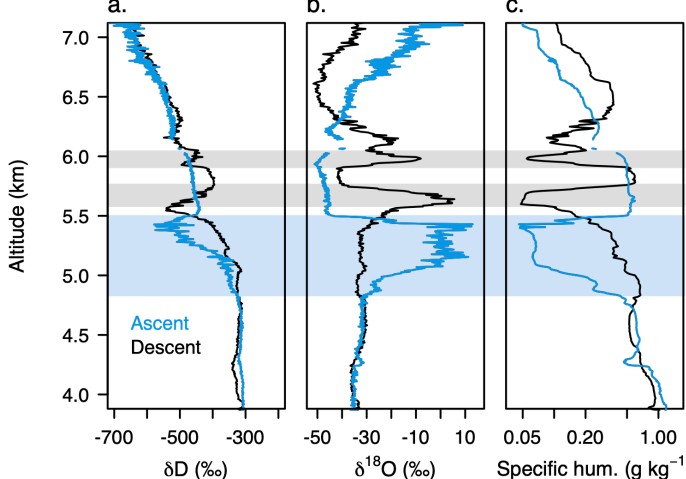


**Figure 9: Strong hysteresis causes (a) the δD vertical profile to differ substantially from vertical profiles of (b) δ¹⁸O and (c) specific humidity during aircraft ascents (blue) and descents (black) on P-3 Research Flight 8. Three dry layers, in which the delayed response and weaker signal in δD are most evident, are indicated by shading. δ¹⁸O is characterized by a better time response than δD but shows unphysical enrichment at altitude due to a shifting humidity-dependent bias over the course of the campaign.**



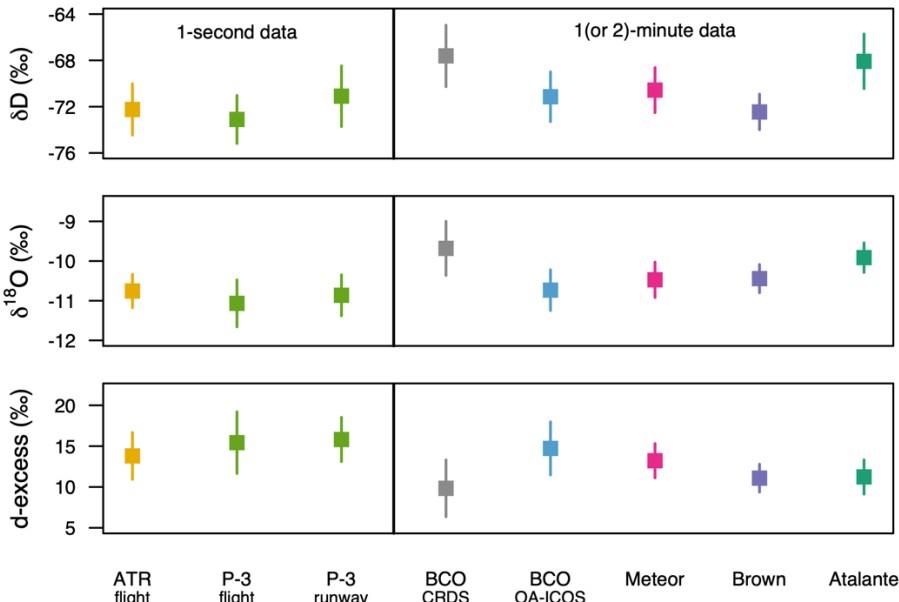

**Figure 10: Campaign-mean near-surface water vapor isotopic values from the various ground, airborne, and ship-based platforms. Whiskers represent standard deviations. Values from in flight represent a height of 150±15 m.a.s.l. With the exception of the BCO OA-ICOS measurements, only data that are not flagged for suspect quality are included in the comparison.**

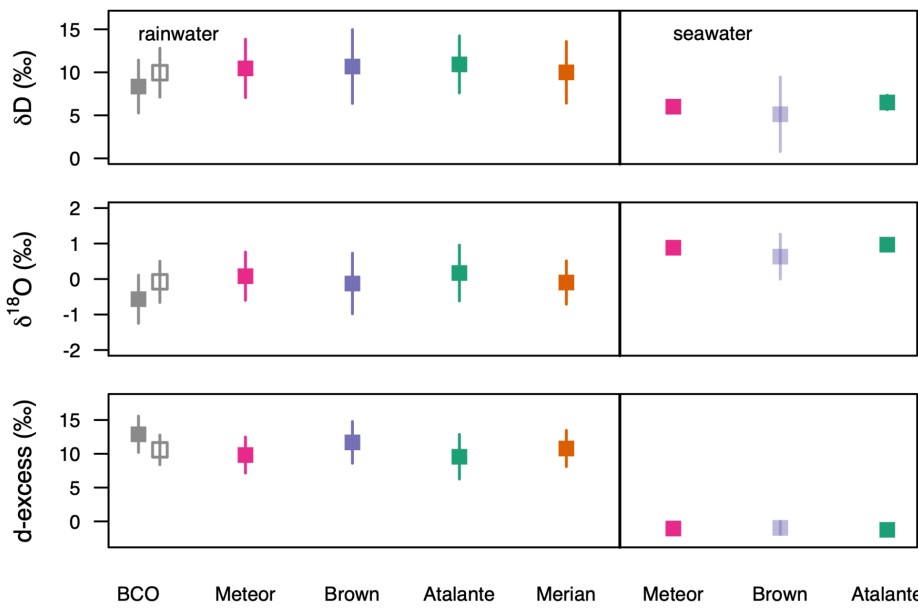

**Figure 11: Campaign-mean rainwater and near-surface seawater values from the BCO and ship-based platforms. Whiskers represent standard deviations. The rainwater comparison excludes Brown samples flagged for delayed collection times, the first Atalante sample, and BCO and Merian samples flagged because no precipitation was detected by the platform's primary meteorological station. Filled symbols for the BCO include rainwater collected every 10**

minutes during the trailing cold front on 22 January while open symbols do not. The seawater comparison excludes flagged Atalante samples and seawater samples taken from a depth greater than 10 m. Brown seawater values are still preliminary and laboratory analysis of Brown seawater ongoing.


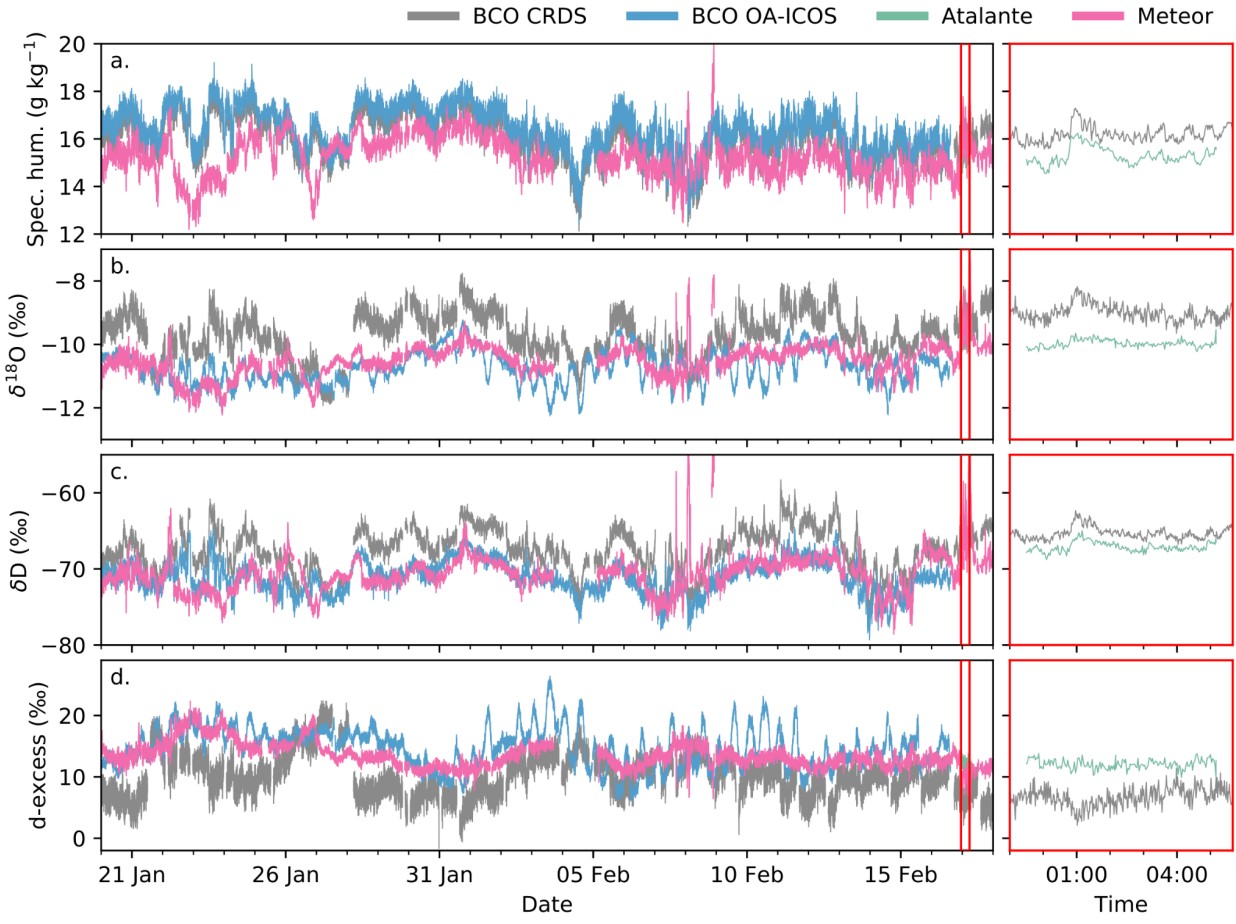


**Figure 12: Time series of (a) water vapor concentration, (b) δ¹⁸O, (c) δD, and (d) *d* from the BCO CRDS analyzer (gray), the BCO OA-ICOS analyzer (blue), and the Meteor (magenta) for the period 20 January to 17 February 2020 (DOY 20-48). The Meteor trace is shifted by 9 hours. Rightmost panels show an enlarged view of the BCO CRDS system and the Atalante (teal) for the period 23:30 16 Feb - 05:10 17 Feb UTC (DOY 47-48), when the Atalante was 2-6 km northeast of the BCO. The Atalante trace is shifted by 10 minutes. (Note that the time series shown here are not smoothed as they are for the comparisons described in Table 2.)**


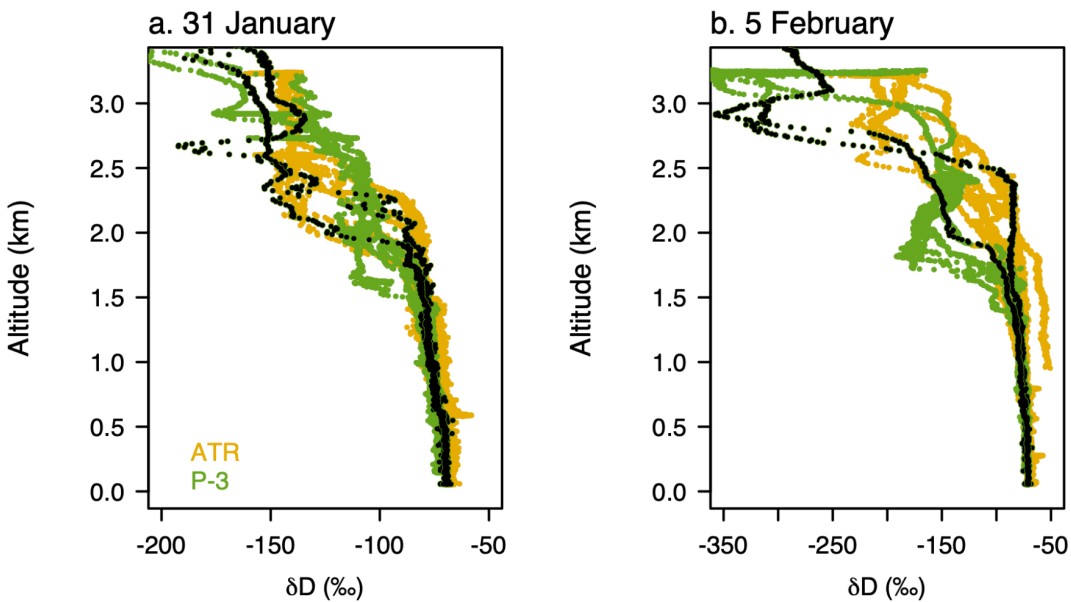

**Figure 13: Vertical δD profiles from the ATR (yellow) and the P-3 (green) for (a) 31 January (DOY 31) and (b) 5 February (DOY 36), the two days on which both aircraft were in the air simultaneously. P-3 observations with black centers represent takeoffs and landings, which were flown in closest proximity to the ATR and show the greatest structural similarity to ATR δD profiles.**

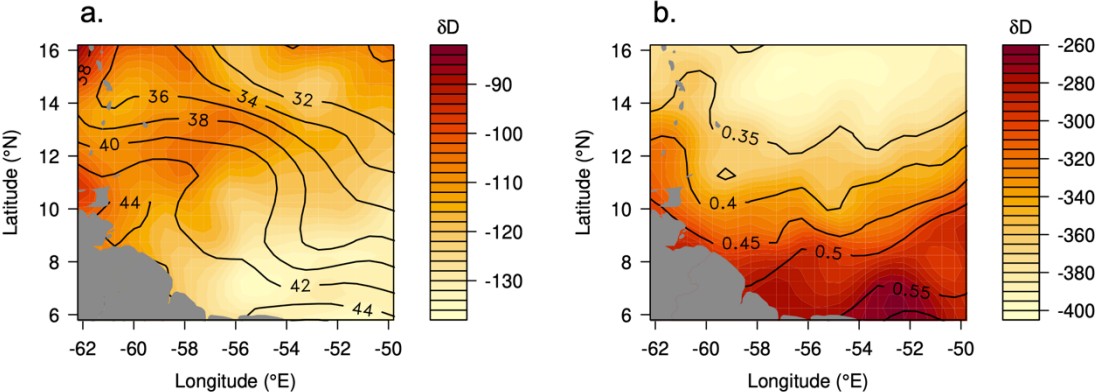

**Figure 14: a) TROPOMI total column δD (permil, shading) and total column water (kg m², contours), averaged on a 0.5-degree grid and smoothed for the period 11 Jan - 20 Feb 2020 (DOY 11-51). Only retrievals with a quality value of 0.5 or higher are selected. b) IASI δD (permil, shading) and specific humidity (g kg$^{-1}$; contours) at 6.4 km, averaged on a 0.5-degree grid and smoothed for the same period. (Note that the two water vapor fields represent distinct quantities.) Only data marked "good quality" in terms of spectral fit are used.**

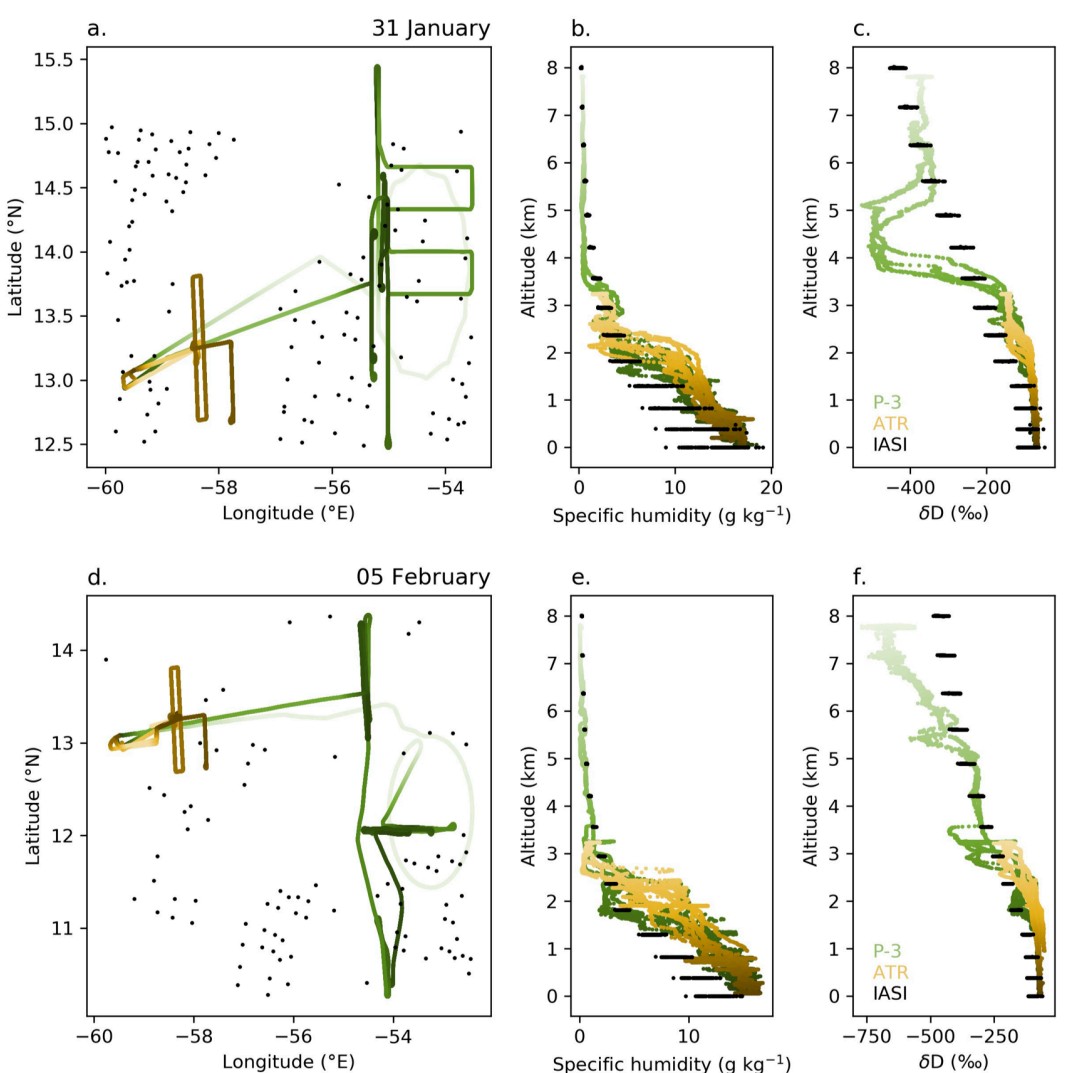

**Figure 15: (a,d) Maps of the measurement locations for IASI (black, morning passes only), the ATR (yellow), and the P-3 (green) on (top) 31 January (DOY 31) and (bottom) 5 February 2020 (DOY 36) and measured vertical profiles of (b,e) specific humidity and (c, f) δD from these days. Note that averaging kernels have not been applied to the aircraft observations to emphasize the different sensitivities of the various observation types.**

