# Peer review of "Isotopic measurements in water vapor, precipitation, and seawater during EUREC4A"

_Earth System Science Data, 2022_

## Referee Comment (RC1)

**Review of « Isotopic measurements in water vapor, precipitation, and seawater during EUREC4A » by Bailey and others.**

This manuscript presents a synthesis of isotopic composition of atmospheric vapour, precipitation and sea water acquired during the EUREC4A campaign, near Barbados. The ambitious scale of the project, including seven vapour infrared spectrometers monitoring the vapour isotopic composition together on land, three ships and two aircrafts, created an invaluable set of timeseries which will be used to study convection in tropical environments.

The manuscript adequately reports the methods and carry out rigorous uncertainty evaluation. The huge dataset leads to a difficult to grasp ensemble of time series, and at times, the manuscript is difficult to follow, especially for people not involved in the project. It seems like the actual use of the produced data might be difficult: indeed, while the authors were careful to provide an exhaustive access to the data, which is commendable, the manuscript describes large part of the dataset that shouldn't be used, or should be used with care, but does not provide a flagged version of the dataset. Considering the size of the dataset, this might be excessively complicated to produce, but probably also reduces the reach of the dataset and the manuscript.

Due to the high quality of the dataset and the post-treatment, validations, and uncertainty evaluation, I recommended to accept the manuscript after the following modifications have been implemented.

**General Comments:**

1. The manuscript really provides a complete overview of everything that has been measured. I agree that including easily accessible calibrated "relatively raw" datasets is great, so other scientists can in the future explore the datasets. But considering that the manuscript provides a detail overview of when the data shouldn't be trusted at face value, a flagged version of the dataset, or alternatively a filtered version of the dataset, would be useful to ensure that other users will not over-interpret data.
2. The general organisation seems relevant, but is sometimes difficult to follow due to the large number of details, and that the analysis and post-treatments were all realised differently, which is how it is. For the sake of using the data, and comparing the different datasets, the same key elements are needed though, regardless of how they were obtained. While the information is clearly provided here, if it was provided in a more systematic manner, it might help the reader.

**Specific Comments:**

Lines 46 to 47: "Moreover, water vapor isotope ratios were measured from a few meters to nearly 8 km above sea level."

While this is true, the quality of the vapour isotopic ratios measured above 5 km might not be sufficient to be used (see comment below). At this point, wouldn't it be fairer to provide the altitude range where you are confident of the trustworthiness of the dataset?

Lines 83 to 85: "As a result, oxygen and hydrogen isotope ratios in vapor (i.e. 18O/16O, D/H) lower with progressive condensation and rainout, while evaporation from the ocean (or another liquid reservoir) and subsequent moisture transport replenish the atmosphere with relatively heavy water. (Although the evaporative process itself discriminates against heavy water, the ocean is isotopically enriched relative to the atmosphere.)"

This is a very complicated way to say that the vapour is more depleted than the ocean water, and gets more depleted in heavy isotope each time a precipitation event occurs.

Lines 86 to 87: "One outcome is the ability to differentiate boundary layer and free tropospheric air and to quantify the exchange of moisture between these layers. "

Can you really differentiate these solely based on the isotopic composition?

Lines 86 to 94: For this whole part of the paragraph, I would say that the description is a little bit optimistic. Water isotopic composition is an integrated variable that is sensitive to all the phase transitions, as well as to diffusion and mixing of air masses, which means that under some assumptions, you would be able to test hypothesis about the history of the air masses, and in particular, which processes to moistening them.

Lines 99 to 100: "We will also use the deuterium excess parameter, defined as d = δD - 8×δ18O, to describe variations in one isotope ratio relative to the other."

The definition of d-excess seems limited. Why 5 then? I guess maybe considering changing for "To describe relative variations of both isotopes which do not follow the "meteoric water line"."

Lines 104 to 105: "A total of seven water vapor isotopic analyzers, sampling at 0.5 Hz or faster, were deployed during the campaign on two research aircraft"

The 0.5 Hz sampling rate seems peculiar and an explanation of why could this be important is probably needed. I presume it is for the processing of the aircraft measurements, but clearly, when seeing the performances of the BCO infrared spectrometers, from the same inlet, at high sampling rate, it seems like there is very limited "climatic interpretation" for data with 0.5 Hz sampling rate.

Overall, it seems like there is a dichotomy between the measurements which took place in certain conditions, and the data which are meaningful only in subset of the range of observed conditions. In the case of the resolution of the produced data, it seems like the recommendation would be to use them at a resolution of several minutes or an hour?

Lines 110 to 112: "This wealth of observational data will aid interpretation of the isotopic signals, just as the isotopic information will provide a new lens through which to evaluate microphysical and dynamical controls on trade wind cloudiness."

Totally agree with this, which is why tuning down the previous paragraph of the introduction makes sense: isotopic signals are very complicated to interpret on their own since it's integrating a potentially important number of processes, and thus they are very useful with hypothesis to test/other observations.

Lines 143 to 144: "The BCO water vapor isotopic measurements were set up to serve as a high-frequency (1 minute) reference dataset at a location with extensive meteorological in situ and remote sensing observations,"

I agree with the statement that 1 minute is high frequency measurement in the case of atmospheric boundary layer dynamic. But then, this is very slow compared to the 0.5 Hz mentioned in the introduction.

Lines 424 to 425: The uncertainties given here are very surprising, in particular compared to the values given lines 413 to 414. Is this due to the rather old model of the Picarro analyser?

Lines 431 to 432: "The effect of this oscillation is not included in the OA-ICOS analyzer's isotopic uncertainty estimates."

If the effect of an oscillation that seems to be an artefact is not included in the uncertainty estimates, the datapoints should be flagged out.

Lines 490 to 495: This is very confusing. Wouldn't the difference between the correction functions obtained in the field vs the ones derived post campaign be related to accuracy while the standard deviation at 1Hz be linked with precision?

Lines 679 to 684: "Surprisingly, the BCO's two analyzers are 1.5 and 4.5‰ different in δ18O and δD, respectively, even though they sampled from the same inlet and were calibrated using the same standards and procedure. This unexpected discrepancy highlights the challenge of accurately estimating biases and uncertainties in water vapor isotopic field measurements using typical calibration approaches and suggests it may be necessary to measure a larger number of standards or to measure the standards for longer periods while in the field."

Isn't these results contradicting with the uncertainty propagation? These values are larger than the ones shown in section 3.1, and thus, would suggest that the uncertainties are clearly underestimated. This is furthermore problematic that these values are of the same order of magnitude than the daily and synoptic variations.

Lines 686 to 689: "Despite their sizable mean offsets, time series from the two BCO analyzers are strongly correlated for both water vapor concentration and δD, bolstering our confidence in the variability captured in their respective signals (Fig. 11, Table 2). (Correlation between the δ18O time series is diminished by the oscillation in the OA-ICOS signal but can be increased by applying a low pass filter or averaging to longer time steps.)"

The correlation of 0.94 between the dD of the BCO CRDS and BCO OA-ICOS appears at odds with Figure 11. Since the dataset stored at Galewsky, J. (2020). BCO LGR Water Vapor Isotopic Composition. [Dataset]. Aeris. https://doi.org/10.25326/309 is faulty, it was impossible to reproduce the calculation. Did you evaluate the correlation on hourly resolved data ?

If you compare to (Leroy - Dos Santos et al., 2020), where two instruments were measuring from two different inlets, and located at neighbouring site, almost 4km away, and with a difference of 470m of altitude, the correlations are larger than 0.9 for humidity and d18O. In your case, it with such large difference for both instruments measuring from the same inlet, either one or both are faulty. It is essential to investigate which of the two datasets is to be trusted, and potentially at which resolution, because clearly here, the 0.5 Hz data do not contain climatic signal.

The correlation between the BCO OA-ICOS and the meteor (and with the other available datasets when they were nearby) could be useful, because it appears that the agreement is actually better for both dD and dexcess than between BCO CRDS and BCO OA-ICOS, excluded the periods with the weird d18O oscillations.

This part is a key weakness of the manuscript, and would really justify an additional effort to provide a safe dataset.

**Bibliography**

Leroy - Dos Santos, C., Masson-Delmotte, V., Casado, M., Fourré, E., Steen-Larsen, H. C., Maturilli, M., Orsi, A., Berchet, A., Cattani, O., & Minster, B. (2020). A 4.5 year-long record of Svalbard water vapor isotopic composition documents winter air mass origin. *Journal of Geophysical Research: Atmospheres*, e2020JD032681.

---

## Author Comment (AC1)

We thank both reviewers for their valuable comments, which we have worked to address below to improve the manuscript. The reviewers' comments are reproduced in blue and our answers are given in black.

**Reviewer 1: Mathieu Casado**

This manuscript presents a synthesis of isotopic composition of atmospheric vapour, precipitation and sea water acquired during the EUREC$^4$A campaign, near Barbados. The ambitious scale of the project, including seven vapour infrared spectrometers monitoring the vapour isotopic composition together on land, three ships and two aircrafts, created an invaluable set of timeseries which will be used to study convection in tropical environments.

The manuscript adequately reports the methods and carry out rigorous uncertainty evaluation. The huge dataset leads to a difficult to grasp ensemble of time series, and at times, the manuscript is difficult to follow, especially for people not involved in the project. It seems like the actual use of the produced data might be difficult: indeed, while the authors were careful to provide an exhaustive access to the data, which is commendable, the manuscript describes large part of the dataset that shouldn't be used, or should be used with care, but does not provide a flagged version of the dataset. Considering the size of the dataset, this might be excessively complicated to produce, but probably also reduces the reach of the dataset and the manuscript. Due to the high quality of the dataset and the post-treatment, validations, and uncertainty evaluation, I recommended to accept the manuscript after the following modifications have been implemented.

General Comments:

1. The manuscript really provides a complete overview of everything that has been measured. I agree that including easily accessible calibrated "relatively raw" datasets is great, so other scientists can in the future explore the datasets. But considering that the manuscript provides a detail overview of when the data shouldn't be trusted at face value, a flagged version of the dataset, or alternatively a filtered version of the dataset, would be useful to ensure that other users will not over-interpret data.

Response: All of the EUREC$^4$A-iso datasets contain uncertainty estimates, which quantify our confidence in the measurements, quality control flags, and/or global quality control comments. Revisions to Sect. 3 of the main text (see next comment) highlight these dataset elements more clearly.

In addition, Table 3, which provides a summary of all the EUREC$^4$A-iso datasets, has been revised to include information about quality control flagging (pointing readers to where to find the quality control information associated with each dataset more easily) and includes additional notes that highlight the most important data quality concerns.

We suspect the reviewer may have been particularly concerned with the quality of the BCO OA-ICOS dataset and the P-3 data at altitude. Of note, the BCO OA-ICOS dataset includes a README file warning potential data users about the suspected spectroscopic issue with the oxygen isotope ratio. Revisions to the main text now state more clearly that the CRDS dataset is the preferred BCO dataset. For example, Lines 786-789 in the revised text state: "While the BCO OA-ICOS values are similar to near-surface water vapor isotope ratios measured at sea...we are confident that the higher isotope ratios of the BCO CRDS system are more accurate and result from sea spray evaporation associated with wave breaking at Barbados' most windward point".

The P-3 1 s data files already contain two estimates of uncertainty (see Fig. 8 in the revision), both of which clearly show that measurement precision declines with decreasing water vapor concentration. Indeed, this is characteristic of all CRDS water isotopic systems (see, for example, the ATR analyzer uncertainties in Fig. 7 of the revision). Based on the perceived concerns of the reviewer, we have opted to include variable-specific comments in the P-3 isotopic data files that warn data users about the general tendency for isotopic measurement precision and accuracy to decline with lower water vapor concentrations and lower isotope ratios. These comments will appear in version 1.2 of the P-3 isotopic data.

Note that, unlike the high-altitude $\delta^{18}O$ data from the P-3, which are flagged for being unphysical, the high-altitude $\delta D$ data are scientifically useful and show meaningful qualitative differences flight-to-flight despite their high uncertainties.

> 2. The general organisation seems relevant, but is sometimes difficult to follow due to the large number of details, and that the analysis and post-treatments were all realised differently, which is how it is. For the sake of using the data, and comparing the different datasets, the same key elements are needed though, regardless of how they were obtained. While the information is clearly provided here, if it was provided in a more systematic manner, it might help the reader.

Response: We have significantly revised Sects. 2-3 such that the same key elements are discussed in the same order for all platforms. For example, the water vapor data collections (Sect. 2) all now include a brief description of the platform's primary sampling strategy, followed by a streamlined description of the installation and inlet system, followed by a description of in-field calibration protocols. The precipitation data collections now all include information about how many samples were collected and whether samples represent discrete rain events or not. They also include similar details about in-field sample treatment and storage and state the laboratory where isotopic analysis was carried out. (If these details are not known, this has been stated.) The seawater collection descriptions now describe more clearly the similarities and differences in sampling strategies between platforms. Like the rainwater descriptions, they now include similar details about in-field sample treatment and storage and state the laboratory where isotopic analysis was carried out.

Section 3 begins with a revised paragraph that highlights key similarities in how the EUREC⁴A-iso datasets were processed. Sect. 3.1 begins with a new paragraph that describes the format by which post-processing is described specifically for the water vapor isotopic measurements. As stated therein, "for each platform, we discuss how in-field and/or pre- and post-campaign calibrations were used to normalize the data to the VSMOW-SLAP scale. We also describe how instrumental drift was evaluated and whether it required adjustments to the VSMOW-SLAP normalization over the course of the campaign. Moreover, we discuss the relevance of known biases associated with low water vapor concentrations…and report any post-processing of water vapor concentration data or adjustments to timestamps to account for time delays in the measurement systems".

Each post-processing description now also includes a clear statement of measurement uncertainty, describes the time resolution of the final data product, and alerts readers to flags or masks in the datasets.

Furthermore, to help make the EUREC⁴A-iso data collection less "difficult to grasp" and to provide readers with an overview of the water cycle processes one might study using the collection, we have included a new figure (Fig. 1), shown below:

[Figure]

➡ vapor enriching processes

⇨ vapor depleting processes

Additional descriptions of data collection uses can be found in Sect. 6 "Concluding perspective on dataset uses", which has changed little from the original submission.

Specific Comments:

1)  Lines 46 to 47: "Moreover, water vapor isotope ratios were measured from a few meters to nearly 8 km above sea level."
    While this is true, the quality of the vapour isotopic ratios measured above 5 km might not be sufficient to be used (see comment below). At this point, wouldn't it be fairer to provide the altitude range where you are confident of the trustworthiness of the dataset?

Response: The P-3 data from above 5 km are still useful and show meaningful qualitative differences flight-to-flight. Lower confidence in the data collected at these altitudes is characterized by the larger uncertainty estimates provided in the data files (see also Fig. 8 in the revision). Nevertheless, we have generalized the statement in the abstract to indicate that measurements were made to the mid-free troposphere.

2)  Lines 83 to 85: "As a result, oxygen and hydrogen isotope ratios in vapor (i.e. 18O/16O, D/H) lower with progressive condensation and rainout, while evaporation from the ocean (or another liquid reservoir) and subsequent moisture transport replenish the atmosphere with relatively heavy water. (Although the evaporative process itself discriminates against heavy water, the ocean is isotopically enriched relative to the atmosphere.)"
    This is a very complicated way to say that the vapour is more depleted than the ocean water, and gets more depleted in heavy isotope each time a precipitation event occurs.

Response: We have adopted your language suggestions and rephrased as follows: "Because isotope ratios (i.e. $^{18}O/^{16}O$, D/H) are sensitive to the integral of moist processes experienced by an air mass during transport (Gat 1996; Galewsky et al., 2016), they are an ideal tool for assessing the coupling between the circulation at large scales and moist processes at smaller scales (Fig. 1). This sensitivity stems from the fact that isotopically heavy and light water molecules change phase and diffuse at distinct rates, causing the heavier molecules to reside in greater relative abundance in the condensed phase. The result is that the atmosphere is depleted

of heavy isotopes relative to ocean water below and becomes further depleted as condensation and rainout occur. In contrast, evaporation from the ocean, and subsequent upward moisture transport, enriches the atmosphere isotopically (even though the evaporative process itself discriminates against heavy water). Evaporation also causes a shift in the hydrogen isotope ratio relative to the oxygen isotope ratio due to diffusive differences between the heavy isotopologues ($H_2^{18}O$ and HDO) under non-equilibrium conditions".

3) Lines 86 to 87: "One outcome is the ability to differentiate boundary layer and free tropospheric air and to quantify the exchange of moisture between these layers. " Can you really differentiate these solely based on the isotopic composition?

Response: We have modified our claim as follows: "Isotope ratios can thus help differentiate between air masses that have experienced distinct water cycle histories (e.g. Noone et al., 2011; Hurley et al., 2012; Bailey et al., 2013; Aemisegger et al., 2021a) and test hypotheses about the processes responsible for setting air mass humidity and cloud states. Examples include evaluating the roles of air-sea exchange and rain re-evaporation in moistening the atmosphere (Fig. 1; Worden et al., 2007; Benetti et al., 2015; Aemisegger et al., 2015; Risi et al., 2020). As pseudo-conserved tracers, isotope ratios can also help characterize mixing between air masses that are isotopically distinct, such as the boundary layer and free troposphere (e.g. Noone et al., 2011; Bailey et al., 2013; Salmon et al., 2019). Indeed, while the isotopic signature of the free troposphere in subsidence-dominated regions like the trades is set primarily by conditions of last saturation (González et al. 2016; Galewsky and Hurley 2010), the isotopic composition of the boundary layer is largely regulated by air-sea interactions and shallow moist convective processes (Benetti et al. 2015; Risi et al. 2020). Mixing between these atmospheric layers produces predictable variations in the isotope ratio as a function of water vapor concentration (Noone et al. 2011; Noone 2012)".

4) Lines 86 to 94: For this whole part of the paragraph, I would say that the description is a little bit optimistic. Water isotopic composition is an integrated variable that is sensitive to all the phase transitions, as well as to diffusion and mixing of air masses, which means that under some assumptions, you would be able to test hypothesis about the history of the air masses, and in particular, which processes to moistening them.

Response: Please see the response above.

5) Lines 99 to 100: "We will also use the deuterium excess parameter, defined as $d = \delta D - 8 \times \delta 18O$, to describe variations in one isotope ratio relative to the other." The definition of d-excess seems limited. Why 5 then? I guess maybe considering changing for "To describe relative variations of both isotopes which do not follow the "meteoric water line"."

Response: To clarify, we are not defining deuterium excess as variations in one isotope ratio relative to another but simply stating that we use the deuterium excess to characterize these variations. Deuterium excess is defined by the formula provided, for which we have added Dansgaard (1964) as a reference.

6) Lines 104 to 105: "A total of seven water vapor isotopic analyzers, sampling at 0.5 Hz or faster, were deployed during the campaign on two research aircraft"

The 0.5 Hz sampling rate seems peculiar and an explanation of why could this be important is probably needed.

Response: We have emphasized the sampling rate of the sensors in order to provide a sense of the scale of the measurements during the EUREC$^4$A deployment, which is the main purpose of this paragraph. The 0.5 Hz acquisition rate you have flagged corresponds to the older isotopic analyzers deployed during EUREC$^4$A (e.g. the Picarro L2120-i deployed at the BCO). In comparison, the L2130-i instruments deployed on the ATR and P-3 were designed in close exchange with Picarro for dedicated aircraft measurements. They feature increased flow rates of up to 0.6 SLPM and increased data acquisition rates of 1-5 Hz, which are necessary to capture higher frequency signals.

I presume it is for the processing of the aircraft measurements, but clearly, when seeing the performances of the BCO infrared spectrometers, from the same inlet, at high sampling rate, it seems like there is very limited "climatic interpretation" for data with 0.5 Hz sampling rate.

Response: EUREC$^4$A-iso is focused on elucidating processes related to shallow convective cloud formation that are otherwise difficult to evaluate with traditional instrumentation (see new Fig. 1). These processes act at turbulent to synoptic timescales (1 s to 1 day) and at cloud- to meso-scales (10 m to $10^5$ m). For this reason, we feel that providing our data at relatively high time resolutions (1 s to 1 min) is not only warranted but also essential. (Keep in mind that a 1 second average on the aircraft represents a footprint of about 100 m. A 1 minute average would represent a footprint of >7 km!) The focus of EUREC$^4$A-iso also differs dramatically from climatic applications that typically use water isotopic information on hourly, daily, or even monthly timescales.

Overall, it seems like there is a dichotomy between the measurements which took place in certain conditions, and the data which are meaningful only in subset of the range of observed conditions. In the case of the resolution of the produced data, it seems like the recommendation would be to use them at a resolution of several minutes or an hour?

Response: We have provided the non-flight data at 1 or 2 minute resolution and the flight data at 1 s resolution, since these are the resolutions we think are most useful for pairing the isotopic data with other meteorological measurements to study the shallow convective environment of the trade winds. For those desiring additional information, Sect. 2 now includes estimates of measurement response (time delay) for each water vapor isotopic analyzer.

While in some cases, the estimated measurement response time may be slightly larger than the time resolution at which the data are produced (e.g. 10 s v. 1 s for the ATR), consider the fact that many of the major observational facilities customarily report data at a particular time resolution, even if the individual measurements are not fully independent at this frequency. Indeed, most airborne platforms customarily report 1 s data even if the individual data points are not fully independent at this resolution. To facilitate the joint use of the isotopic data with other meteorological data, we have tried, where possible, to produce datasets with a time resolution that matches other data from the same observing platform.

7)    Lines 143 to 144: "The BCO water vapor isotopic measurements were set up to serve as a high-frequency (1 minute) reference dataset at a location with extensive meteorological in situ and remote sensing observations,"

I agree with the statement that 1 minute is high frequency measurement in the case of atmospheric boundary layer dynamic. But then, this is very slow compared to the 0.5 Hz mentioned in the introduction.

Response: Yes, despite the 0.5 Hz data acquisition rate, we recommend using the data at 1 min time resolution, in the case of the BCO datasets, to take advantage of the optimal precision of the data. We have removed "high frequency", since, indeed, in terms of boundary layer turbulence, 1 min is rather slow. We have changed the text as follows: "The BCO water vapor isotopic measurements were set up to serve as a reference dataset at 1 minute time resolution at a location with extensive meteorological in situ and remote sensing observations…"

8)  Lines 424 to 425: The uncertainties given here are very surprising, in particular compared to the values given lines 413 to 414. Is this due to the rather old model of the Picarro analyser?

Response: We realize that there might be a source of confusion here. The indicated precision of the calibration (originally at Lines 413-414; now Line 454) relates to an averaging time window corresponding to the length of the calibration run (thus 10 to 30 minutes). The total uncertainty for the processed isotopic data, indicated in Lines 472-474 of the revision, is also affected by the precision of the measurement at 1 minute time resolution, the drift correction, and the uncertainty associated with the isotopic composition of the liquid standards. For reference, these uncertainties are comparable to what we obtained in previous deployments at the BCO and other sites (Aemisegger et al., 2021; Aemisegger et al., 2014; Aemisegger et al.; 2012). To help clarify, we have added to the text the time resolution of the measurement to which the uncertainty corresponds.

One of the messages we hope to convey in this manuscript is that uncertainty estimates associated with water vapor isotopic sampling are typically much larger than factory-reported precision estimates, largely due to errors in generating reference gas from a liquid standard, in fitting the normalization correction function, and in characterizing biases associated with low water vapor concentration. The EUREC$^4$A water vapor isotopic datasets attempt to account for all of these possible sources of uncertainty.

9)  Lines 431 to 432: "The effect of this oscillation is not included in the OA-ICOS analyzer's isotopic uncertainty estimates."
     If the effect of an oscillation that seems to be an artefact is not included in the uncertainty estimates, the datapoints should be flagged out.

Response: Due to the difficulty of knowing exactly when the oscillation affected measurements or not (and therefore of flagging these points), we have opted instead to report all  BCO OA-ICOS δ$^{18}$O measurements as suspect. This is done in the README file archived with the datafiles and now, based on your concern, also in Table 3.

10) Lines 490 to 495: This is very confusing. Wouldn't the difference between the correction functions obtained in the field vs the ones derived post campaign be related to accuracy while the standard deviation at 1Hz be linked with precision?

Response: Variations in the P-3 normalization functions obtained in the field were the result of our inability to generate reference gas precisely (i.e. with repeated reliability) not due to an inaccuracy (i.e. a bias). That said, given the confusion this paragraph created, we have revised it and amended our recommendation. Because the standard deviations reported in the 1 s P-3 files are about twice the uncertainty estimated from calibration checks, we recommend data users use the standard deviations as a more conservative estimate of total uncertainty.

11) Lines 679 to 684: "Surprisingly, the BCO's two analyzers are 1.5 and 4.5‰ different in δ18O and δD, respectively, even though they sampled from the same inlet and were calibrated using the same standards and procedure. This unexpected discrepancy highlights the challenge of accurately estimating biases and uncertainties in water vapor isotopic field measurements using typical calibration approaches and suggests it may be necessary to measure a larger number of standards or to measure the standards for longer periods while in the field."

Isn't these results contradicting with the uncertainty propagation? These values are larger than the ones shown in section 3.1, and thus, would suggest that the uncertainties are clearly underestimated. This is furthermore problematic that these values are of the same order of magnitude than the daily and synoptic variations.

Response: We agree that uncertainties estimated from traditional calibration procedures are likely underestimated, and we have made this point throughout the text. Based on continued exploration of the data, we believe the source of the discrepancy most likely stems from a water vapor concentration dependency in OA-ICOS analyzers that is apparent even at high humidity levels (cf., Sturm and Knohl, 2010). We have newly noted this possibility in Sect. 3.1 and 4.1.1 (e.g., Lines 789-791 of the revision state: "The lower isotope ratios of the BCO's OA-ICOS analyzer likely reflect an uncorrected water vapor concentration bias that can be significant for OA-ICOS systems even at high humidity levels (Sturm and Knohl, 2010)"). Unfortunately, due to problems with the calibration system at the BCO we could not perform the necessary water vapor mixing ratio dependency tests in the field to characterize the humidity dependence of the OA-ICOS analyzer deployed and bias-correct for it.

We believe the "missing" humidity dependent correction manifests itself as a humidity-dependent difference between the CRDS and OA-ICOS data (see new Fig. S3-S4 below, which have been added to the Supplemental Information). From a previous deployment of the same OA-ICOS system on the Azores, we estimate that the amplitude of such a correction (Galewsky, 2021, https://www.arm.gov/publications/programdocs/doe-sc-arm-19-027.pdf) could lead to a shift in δD of about 3‰ and a shift in δ$^{18}$O of about 1‰ at the humidity levels measured at the BCO (this is now stated in the revisions to Sect. 4.1.1). Note that adjustments of this magnitude would bring the OA-ICOS time series to within the uncertainty band of the CRDS time series (see new Fig. S4-S5 below, which have been added to the Supplemental Information).

[Figure]

Figure S3: OA-ICOS-CRDS differences in δD (y-axis) as a function of the OA-ICOS water vapor concentration (x-axis). Shading shows the OA-ICOS (normalized and drift-corrected) δD value in units ‰.

[Figure]

Figure S4: The scatterplot shows the original relationship between the (normalized and drift-corrected) OA-ICOS and CRDS δD values (blue dots). It also shows how the relationship shifts towards the 1:1 line (black, solid) when the OA-ICOS data are scaled by one of three methods: a simple offset (green dots), a simple linear regression (red dashed line, gold dots), or a total least squares regression (black dashed line, orange dots).

[Figure]

Figure S5: After scaling the original OA-ICOS data (light blue dots) to the CRDS data by one of the methods illustrated in Fig. S4, the adjusted δD (‰) time series (green, gold, or orange dots) converges with the CRDS δD time series (dark blue dots).

Despite differences in the absolute values between the CRDS and OA-ICOS systems at the BCO, we insist that there are very high correlations between the signals measured by the two instruments for $q$ and δD. The artificial oscillations in the $\delta^{18}O$ measured by the OA-ICOS prevent us from using the oxygen isotope ratios in a similar comparison.

12) Lines 686 to 689: "Despite their sizable mean offsets, time series from the two BCO analyzers are strongly correlated for both water vapor concentration and δD, bolstering our confidence in the variability captured in their respective signals (Fig. 11, Table 2). (Correlation between the δ18O time series is diminished by the oscillation in the OA-ICOS signal but can be increased by applying a low pass filter or averaging to longer time steps.)" The correlation of 0.94 between the dD of the BCO CRDS and BCO OA-ICOS appears at odds with Figure 11. Since the dataset stored at Galewsky, J. (2020). BCO OA-ICOS Water Vapor Isotopic Composition. [Dataset]. Aeris. https://doi.org/10.25326/309 is faulty, it was impossible to reproduce the calculation. Did you evaluate the correlation on hourly resolved data ?

Response: We do not understand why the correlations reported in Table 2 (which represent 1 hourly smoothed data with 1 minute resolution) are at odds with the BCO time series comparison figure (Fig. 12 in the revision). As stated above, despite differences in the absolute values between the CRDS and OA-ICOS systems at the BCO, both analyzers capture the same environmental variability; hence, their high correlations.

Below in Fig. R1 we show the correlation between the CRDS and the OA-ICOS signals for different averaging windows.

[Figure]

Figure R1: Pearson correlations for different time windows (aggregation in minutes, x-axis) for the specific humidity $q$, δD and $δ^{18}O$ (y-axes) measured by the Picarro CRDS system and the LGR OA-ICOS system at the BCO.

If you compare to (Leroy - Dos Santos et al., 2020), where two instruments were measuring from two different inlets, and located at neighbouring site, almost 4km away, and with a difference of 470m of altitude, the correlations are larger than 0.9 for humidity and d18O. In your case, it with such large difference for both instruments measuring from the same inlet, either one or both are faulty.

Response: We believe that there is a misunderstanding here: the difference between the CRDS and OA-ICOS time series in absolute values and the correlations are two independent measures of agreement. Yes, we do observe an offset between the CRDS and the OA-ICOS time series at the BCO, which, as explained above, we believe is due to a bias introduced in the OA-ICOS system because of the missing water vapor concentration correction. However the correlation between the two instruments is excellent for $q$ and δD as can be seen in new Fig. S5 above.

Furthermore, please note that the BCO comparison is between two laser spectrometers that use a different technology and different absorption peaks (one OA-ICOS system and one CRDS system). We now draw readers' attention to this fact at the end of the first paragraph of Section 2.1.1: "The two systems operate at different wavelengths in the infrared; consequently, baseline effects due to varying water vapor concentrations can affect the measurements differently (Johnson and Rella, 2017)".

Comparing two CRDS analyzers (such as in Leroy - Dos Santos et al., 2020) is different because it can potentially mask biasing effects that affect the analyzers in the same way, such as baseline effects due to the presence of other gases (Johnson and Rella, 2017).

> It is essential to investigate which of the two datasets is to be trusted, and potentially at which resolution, because clearly here, the 0.5 Hz data do not contain climatic signal.

Response: We agree and have worked to provide more clarity in this regard in the revised version, emphasizing the trustworthiness of the CRDS data. That said, we would like to point out that there is much to be learned about shallow cumulus cloud formation – from the timescale of individual updrafts to transitions between different cloud patterns at the synoptic timescale – from the BCO's isotopic variability. The fact that the two isotopic analyzers at the BCO agree so well in their ability to capture this variability gives us high confidence that we are able to resolve the shallow convective processes we are interested in. This is one reason we feel that making both datasets publicly available is valuable.

To further emphasize the strong coherence between the two BCO analyzers, new Fig. S4 (replicated above) shows a scatter plot relating the OA-ICOS and CRDS data and illustrates how correcting the OA-ICOS data towards the CRDS data with different methods (e.g. mean offset shift, linear model correction, total least squares correction) leads to an agreement of the two signals within their uncertainty range. New Fig. S5 shows the time series corresponding to these corrections. Both figures have been added to the Supplemental Information.

> The correlation between the BCO OA-ICOS and the meteor (and with the other available datasets when they were nearby) could be useful, because it appears that the agreement is actually better for both dD and dexcess than between BCO CRDS and BCO OA-ICOS, excluded the periods with the weird d18O oscillations.

Response: The correlations between the OA-ICOS and the instruments on the ships are similar to the ones between the CRDS and the ships (given the very high correlations between the OA-ICOS and the CRDS; Table 2). The offset between the OA-ICOS and the ships is indeed a bit less than between the CRDS and the ships. But given the missing water vapor mixing ratio correction in the OA-ICOS data, this might be a coincidence. We attribute the more enriched isotope signals from the CRDS at the BCO to the contribution of sea spray evaporation from waves breaking on the reef and at the cliff just in front of the BCO. This contribution was likely much smaller on the ships. We have added this explanation to Section 4.1.1.

> This part is a key weakness of the manuscript, and would really justify an additional effort to provide a safe dataset.

Response: Strong agreement in absolute value between the two BCO analyzers would, of course, have been preferred. However, their disagreement presents a rare opportunity to highlight the challenges associated with characterizing and correcting for biases and estimating related uncertainties. As mentioned above, the revised text (Sect. 4.1.1) now clearly encourages data users to trust the absolute values of the CRDS system over the OA-ICOS one.

**Reviewer 2:**

The manuscript presents the data collected during the EUREC4A-iso measurements, a sub-part of the EURA4C field campaign dedicated to isotopic measurements.

I find this manuscript very well written, with clear depictions of the various isotopic measurements. I do not have strong concerns about this manuscript, and believe that this manuscript could be published after the really minor comments below have been taken into consideration.
1) Remark : the unit "nmi" is used for distance, but is non-SI. A SI unit should be used.

Response: We have replaced nautical miles with distance estimates in km everywhere.

2) Line 85-90 : I do not understand how isotope ratios can help differentiate boundary layer air and free tropospheric air. It seems to be a shortcut, but this shortcut is not straightforward. Please explain.

Response: As noted above, we have modified the main text as follows: "Isotope ratios can thus help differentiate between air masses that have experienced distinct water cycle histories (e.g. Noone et al., 2011; Hurley et al., 2012; Bailey et al., 2013; Aemisegger et al., 2021a) and test hypotheses about the processes responsible for setting air mass humidity and cloud states. Examples include evaluating the roles of air-sea exchange and rain re-evaporation in moistening the atmosphere (Fig. 1; Worden et al., 2007; Benetti et al., 2015; Aemisegger et al., 2015; Risi et al., 2020). As pseudo-conserved tracers, isotope ratios can also help characterize mixing between air masses that are isotopically distinct, such as the boundary layer and free troposphere (e.g. Noone et al., 2011; Bailey et al., 2013; Salmon et al., 2019). Indeed, while the isotopic signature of the free troposphere in subsidence-dominated regions like the trades is set primarily by conditions of last saturation (González et al. 2016; Galewsky and Hurley 2010), the isotopic composition of the boundary layer is largely regulated by air-sea interactions and shallow moist convective processes (Benetti et al. 2015; Risi et al. 2020). Mixing between these atmospheric layers produces predictable variations in the isotope ratio as a function of water vapor concentration (Noone et al. 2011; Noone 2012)".

3) Lines 660-665 : Does this part refers to Figures 9 and 10 ? The titles of these 2 figures (9 and 10) only states "campaign-mean (…)". The 3 vessels (Atalante, Meteor and Brown) didn't have the same legs, so I wonder how campaign-means can be reasonably used to look at the consistency between the measurements : there are differences in the length of the measurement period, and there are differences in the areas that have been sampled. Can you comment ?

Response: Indeed the platforms did not sample at the same location over the same time window, which implies that natural variability as well as measurement uncertainties contribute to the differences observed in the campaign means across platforms. Nevertheless, showing and intercomparing campaign means is a sensible way to present a large amount of data in a compact way. We have revised the beginning of Section 4.1.1 as follows: "given the integrative nature of water vapor isotope ratios (e.g. Moerman et al., 2013) and the relatively long duration each analyzer sampled, we expect average isotopic differences across platforms to be dominated by spatial variability".

4) Note : Figure 10 is not referenced in the text.

Response: Thank you for catching this typo. You will now find the first reference to original Fig. 10 (Fig. 11 in the revision) on line 767. Additional information about the figure is now included in the paragraph beginning on Line 775 in the revision: "Cross-platform coherence in rainwater improves further if the BCO samples from the trailing cold front (22 January; DOY 22) are also excluded from the campaign-mean averages (open symbols; Fig. 11). Because rain on the 22

January was associated with large-scale convergence, its isotope ratios are much lower than samples representing typical shallow convective showers…"

5) Figure 13: the unit of dD is missing.

Response: The unit has been added to the caption.

**Additional modifications**

Additional modifications have been made to the main text and Supplemental Information based on continued analysis of the data. In particular, we have found that high isotope ratio spikes in the Meteor water vapor isotopic time series may coincide with cold pool processes rather than evaporation of moisture from the ship's decks and surfaces (see Lines 618-622 in the revision and revised Fig. S7).

BCO precipitation samples do not show evidence of post-sampling evaporation, as we had first suspected (see revisions to Sect. 3.2). Instead, we find that frontal rain appears to be isotopically distinct from shallow convective showers due to differences in rain formation and post-condensational exchange processes (see new paragraph starting at Line 775). We have highlighted this by adding a new open symbol to Fig. 11 in the revision.

Table 2 numbers have been updated using the latest versions of the published datasets with 1 hourly smoothing consistently applied.

The BCO time series comparison (Fig. 12 in the revision) has also been updated with the latest versions of the published data.

The near-surface water vapor values shown in Fig. 10 in the revision and the map of the sampling tracks (Fig. 2 in the revision) have been updated to reflect the latest published revision of the Meteor dataset.

The timeline plot (Fig. 3 in the revision) has been updated to include an Atalante rainwater sample that was missing from the original figure.

The paragraph concluding Sect. 4.2 has been modified for technical accuracy.

The water vapor correction function for the P-3 in the Supplemental Information (Eq. S3) has been fixed for a typo in the original submission.

Finally, we have updated citations, fixed a few grammatical errors, and made a few minor stylistic changes.

Thank you for your time and consideration.

**References**

Aemisegger, F., Vogel, R., Graf, P., Dahinden, F., Villiger, L., Jansen, F., Bony, S., Stevens, B., and Wernli, H.: How Rossby wave breaking modulates the water cycle in the North Atlantic trade wind region, Weather Clim. Dynam., 2, 281–309, https://doi.org/10.5194/wcd-2-281-2021, 2021.

Aemisegger, F., Sturm, P., Graf, P., Sodemann, H., Pfahl, S., Knohl, A., and Wernli, H.: Measuring variations of δ18O and δ2H in atmospheric water vapour using two commercial laser-based spectrometers: an instrument characterisation study, Atmos. Meas. Tech., 5, 1491–1511, https://doi.org/10.5194/amt-5-1491-2012, 2012.

Aemisegger, F., Pfahl, S., Sodemann, H., Lehner, I., Seneviratne, S. I., and Wernli, H.: Deuterium excess as a proxy for continental moisture recycling and plant transpiration, Atmos. Chem. Phys., 14, 4029–4054, https://doi.org/10.5194/acp-14-4029-2014, 2014.

Galewsky, J.: Water Vapor Isotopic Composition from the Azores Field Campaign Report,DOE/SC-ARM-19-027, https://www.arm.gov/publications/programdocs/doe-sc-arm-19-027.pdf, 2021.

González, Y., Schneider, M., Dyroff, C., Rodríguez, S., Christner, E., García, O. E., Cuevas, E., Bustos, J. J., Ramos, R., Guirado-Fuentes, C., Barthlott, S., Wiegele, A., and Sepúlveda, E.: Detecting moisture transport pathways to the subtropical North Atlantic free troposphere using paired H2O-δD in situ measurements, Atmos. Chem. Phys., 16, 4251–4269, https://doi.org/10.5194/acp-16-4251-2016, 2016.

Johnson, J. E. and Rella, C. W.: Effects of variation in background mixing ratios of N2, O2, and Ar on the measurement of δ18O–H2O and δ2H–H2O values by cavity ring-down spectroscopy, Atmos. Meas. Tech., 10, 3073–3091, https://doi.org/10.5194/amt-10-3073-2017, 2017.

Noone, D. Pairing Measurements of the Water Vapor Isotope Ratio with Humidity to Deduce Atmospheric Moistening and Dehydration in the Tropical Midtroposphere, Journal of Climate, 25(13), 4476-4494, 2012.

Sturm, P. and Knohl, A.: Water vapor δ2H and δ18O measurements using off-axis integrated cavity output spectroscopy, Atmos. Meas. Tech., 3, 67–77, https://doi.org/10.5194/amt-3-67-2010, 2010.

---

## Editor Decision (ED1)

Comment ESSD-2022-03

Need to measure water vapor accurately - a hard-enough problem - then determine isotopic composition which adds substantial additional challenges. Authors apparently have skipped first step, going directly to the second step. But, the first step provides very necessary constraints on instrument performance? E.g laser-based instruments - according to manufacturers - lose precision at high humidities? These authors tend to focus on low humidities more typical of natural environments but still minimize those effects? Dismissed here as so-called 'baseline effects' (line 156).

Authors present data compiled from ground-, air-, ship-, and lab-based systems using platform-specific inlets, standard or customized instruments of varying reliability, response time and sensitivity, calibrated (or, not) through independently-determined procedures. None of which they control or even influence! Remarkable effort to even present such an assemblage; good on them for the effort. Strongly agree with one of their summary sentences, e.g. from line 793 "challenge of accurately characterizing and correcting for all relevant biases in field-deployable water vapor isotopic instruments". This reader thinks they provide very skillful assessments; who if anyone could have handled and described such a variety of data. ESSD should publish careful data compilation efforts regardless of 'success' or 'failure' which mostly derive from needs and interests of users. Authors could/should make better presentation!

Basically, we need cautions and overall disclaimers upfront, e.g. in Abstract. Even to finish with a sentence or two about necessary cautions in many uses of these data? As now presented, the abstract offers only deployment summaries, e.g numbers of instruments on which platform. Substantial uncertainties and cautions only emerge in section 4. Give readers an earlier hint?

Line 47 conflicts with line 45. E.g. if ships collected seawater (line 45, also at line 64) then vertical extent of samples can not start "a few meters above sea level". Perhaps for vapor phase but not for all isotopic samples?

ESSD will require DOI information (referred here to Section 5) repeated at end of abstract. Section 5 unfortunately reports URLs (unreliable), not DOIs (reliable). Table 3, with individual data sets references by DOI seems, again unfortunately, incomplete. Or, incompletely documented. User needs easy access to full set of products. Either convert one of the AERIS or NCEI links to a DOI labelled product or put all products together under a third-party archive service (e.g. Zenodo?) which will provide top-level encompassing DOI, reliable off-site storage, plus very good version control. Not acceptable in current form or format.

Line 95: "last" I think you mean 'most recent'?

Line. 118: "new" I think you mean additional, especially because you have just expended several sentences to justify isotopic measurements based on past data.

Section 2: not clearly specified, perhaps will clarify later, but I suspect:

7 vapor-phase measurements: 2 at BCO, 1 on ATR, 1 on P-3, 3 on ships, sum = 7?

5(?) liquid-phase (precip) measurements: 1 at BCO, 3 on ships above, 1 on additional ship = 5?

3 seawater samplers on 3 ships = 3?

From Fig 2, rainwater samples occurred mostly along longitude approx 57W, with relatively few exceptions.

Line 145: very strong statement here: "… no island effects …". No upwind island effects? This needs referencing to back it up?

Line 148, "regionally representative": I think you mean representative across tropical trade wind environments globally but as written readers could interpret statement as referring only to local BCO environment? Needs some revision.

Line 198, "stored at room temperature": this reader doubts that authors could keep samples at steady temperature during long-range (BCO to Freiburg lab) long-duration transport and storage. Perhaps not important for isotopes? Need some justification here?

Line 219: that the Picarro instrument worked (acceptably?) during deployment nearly a decade earlier provides little help here about reliability, precision, accuracy?

Line 222: please give temperatures consistently in K or C, or explain why one seems appropriate in some cases but not others? Here readers confront inlet reference temperatures in K followed - within same sentence - by tubing heating temperatures in C! One or the other? Not both unless justified for valid reasons.

Line 292: text here, e.g. "limit particle debris" bears (too) remarkable similarity to earlier text describing aircraft inlets (e.g. line 224). One suspects authors adopted text from ship and aircraft sources - fair enough - and no doubt particulate contamination proves troublesome in both cases but authors either need to acknowledge similarity of text (and sampling challenge) or take more care about repeating identical phrases.

Line 294: "sniff tests"? Sloppy, at best. Sniff for RH changes? For odiferous tracers? Certainly not for isotopic composition! Again one suspects authors repeat text from ship-board operators / data providers but seems seriously out-of-place in a careful presentation of isotope data.

Line 427, Section 3 on data processing expends several sentences on water vapor quantification, both as mole fraction and as humidity. Surprised because not addressed in measurement sections?

Line 439: "water vapor isotopic measurements varied widely". Okay, not really a surprise, but have authors given us tools to adjust expectations and make use? For this reader, no.

Side-by-side systems at BCO should have provided "reference dataset with 1 minute time resolution". Failed. Authors conclude (line 793): "unexpected discrepancy between the BCO analyzers". Comparable systems on ATR and P-3 also failed intercomparison critera. Data not bad from ATR (albeit with uncertain response times and corrections) but not - unfortunately - comparable to P-3 data which, apparently, only reference to prior P-3 data. Comparable systems on ships? Only barely useful. Authors know these discrepancies better than any reader but fail to provide a coherent summary. If they can't extract useful summary, who can?

Authors informed assessments of data utility (e.g. Section 4):

For vapor phase, perhaps a "subtle" (or, later, "very subtle) latitudinal pattern emerges. This reader can neither see nor credit such spatial patterns but accepts that authors and other users might. These cautions need to move earlier, e.g. in abstract?

Storage effects, which vary from 'frozen' to even 'poisoned then frozen' to room temperature and even 'subject to drastic heating', impose evaporation and biological effects on isotope

ratios. Authors recognize such issues and provide (where possible) summaries of storage protocols but only rarely deal with the larger issue? E.g. Brown samples should differ substantially from Meteor samples due to differences in storage? Systematic or erratic? Not clear and not well addressed?

Line 1425: Given the scale of aircraft tracks, land mass (grey) in Figure 2 represents South America with Trinidad/Tobago clearly visible. Barbados (13N, 59W), apparently in black, fails to appear to these old eyes even under extreme zoom. All land masses should stay dark grey, particularly if you want color matching to Fig 3. Show location of Deebles Pt BCO? As readers find later, this scale driven by satellite products while less helpful for immediate locale.

Authors make extensive reference to and use of column-integrated satellite products, on two or more spatial scales. Not surprising given EUREC4A relevance and motivation, and authors have offered good access and reasonable interpretation. Summary (in my words): not bad, nothing in remotely sensed data proves or disproves EUREC4A in situ data but subsequent users should take great care in any such intercomparisons for a variety of reasons. Some such caution should emerge as a clearer outcome?

Because of focus on vertical profiles driven by remote sensing, this reader notes again absence of radiosonde profiles (e.g. BCO must launch sondes daily) and of dropsonde profiles from HALO and P-3. Because EUREC4A expended efforts in track planning and resources in sondes themselves, and because correlation with water vapor / humidity turns out such an important factor in isotope measurements, why have authors not at least mentioned sonde humidity profiles? Even to say 'not useful'. To this readers, seems a strange omission.

Authors use term 'diurnal' when in fact they mean 'diel'. Strictly, diurnal refers to daylight, nocturnal refers to night, diel refers to full 24-hour cycle.

---

## Author Response (AR2)

Comment ESSD-2022-03

*Thank you for this careful read of our paper and for these very constructive comments. Answers to each criticism are addressed below (in italics).*

Need to measure water vapor accurately - a hard-enough problem - then determine isotopic composition which adds substantial additional challenges. Authors apparently have skipped first step, going directly to the second step. But, the first step provides very necessary constraints on instrument performance? E.g laser-based instruments - according to manufacturers - lose precision at high humidities? These authors tend to focus on low humidities more typical of natural environments but still minimize those effects? Dismissed here as so-called 'baseline effects' (line 156).

*To communicate instrument performance specifications and describe more clearly our confidence in the water vapor measurements, we have added a new paragraph, at the top of Section 3.1, that conveys the following information. First, previous lab and field-based studies have demonstrated that the water vapor concentration measurements from the types of laser analyzers deployed during EUREC$^4$A are precise and stable over a large concentration range (e.g. 200 - 30,000 ppmv) and over long periods (e.g. years). Second, the latest version analyzers are now designed to perform optimally in the range 1,000 - 50,000 ppmv. The new paragraph also points readers to other ESSD special issue data papers that provide more detailed comparisons between humidity measurements from the airborne isotopic analyzers and other aircraft sensors, which demonstrate the accuracy of these systems.*

*Furthermore, the effects of any inaccuracies in water vapor concentration on the isotopic measurements are removed when correcting the isotopic data for known biases associated with low water vapor concentrations and normalizing them to the VSMOW-SLAP scale. Section 3 provides extensive details on these corrections and refers readers to early papers on this topic such as Aemisegger et al. (2012) and Bailey et al. (2015).*

Authors present data compiled from ground-, air-, ship-, and lab-based systems using platform-specific inlets, standard or customized instruments of varying reliability, response time and sensitivity, calibrated (or, not) through independently-determined procedures. None of which they control or even influence! Remarkable effort to even present such an assemblage; good on them for the effort. Strongly agree with one of their summary sentences, e.g. from line 793 "challenge of accurately characterizing and correcting for all relevant biases in field deployable water vapor isotopic instruments". This reader thinks they provide very skillful assessments; who if anyone could have handled and described such a variety of data. ESSD should publish careful data compilation efforts regardless of 'success' or 'failure' which mostly derive from needs and interests of users. Authors could/should make better presentation!

*Thank you for this encouragement. We have reworked the Abstract and Introduction so that they better describe the needs met with this collection of data (namely, closing water isotopic budgets, characterizing fluxes, and evaluating isotopically enabled numerical simulations) while also giving readers more context on data uncertainties and quality flags. Specifically, by providing detailed reports of uncertainties and flagging (rather than masking) data, we hope to promote more accurate cross-platform comparisons, raise the bar for uncertainty reporting within the nascent water vapor isotope measurement community, and facilitate an open dialogue around improving instrument performance, sampling installation, collection and calibration protocols, and, ultimately, data quality. There should be no doubt, however, that EUREC⁴A's isotopic measurements were an enormous success, which our new framing aims to convey more clearly.*

Basically, we need cautions and overall disclaimers upfront, e.g. in Abstract. Even to finish with a sentence or two about necessary cautions in many uses of these data? As now presented, the abstract offers only deployment summaries, e.g numbers of instruments on which platform. Substantial uncertainties and cautions only emerge in section 4. Give readers an earlier hint?

*We have reworked the Abstract so that it lists data of quality concern and refers readers to Table 3, Section 5 Data availability for a comprehensive list both of dataset DOIs and dataset flags. If one excludes these flagged data from comparisons, as we have done in Section 4 (except when wanting to highlight discrepancies), data users should be able to merge individual datasets as needed for greater spatiotemporal coverage or for closing water isotopic budgets. The cross-platform comparisons in Section 4, especially the strong coherence illustrated by Figures 10, 11, and 13, further bolster our confidence in this statement.*

Line 47 conflicts with line 45. E.g. if ships collected seawater (line 45, also at line 64) then vertical extent of samples can not start "a few meters above sea level". Perhaps for vapor phase but not for all isotopic samples?

*We have clarified this point by stating that vapor measurements extended from a few meters above sea level to the mid-free troposphere and that seawater samples spanned the ocean surface to several km depth.*

ESSD will require DOI information (referred here to Section 5) repeated at end of abstract. Section 5 unfortunately reports URLs (unreliable), not DOIs (reliable). Table 3, with individual data sets references by DOI seems, again unfortunately, incomplete. Or, incompletely documented. User needs easy access to full set of products. Either convert one of the AERIS or NCEI links to a DOI labelled product or put all products together under a third-party archive service (e.g. Zenodo?) which will provide top-level encompassing DOI, reliable off-site storage, plus very good version control. Not acceptable in current form or format.

*We have followed ESSD special issue papers Quinn et al. (2021) and Bony et al. (2022) in referring to the Data availability section of the manuscript rather than listing individual DOIs in the Abstract. However, we now make reference to Section 5 Data availability at the end of the Abstract (rather than in the middle). If this is insufficient, please let us know, and we will make additional changes.*

*Our decision to provide individual DOIs in Table 3 (Section 5 Data availability) was motivated by ESSD special issue papers Pincus et al. (2021), Quinn et al. (2021), and Bony et al. (2022). Our table provides searchable DOIs for every dataset except the Brown sea samples, which are provided in the Supplemental. We have opted to continue providing data users with individual dataset DOIs while also requesting an umbrella DOI, as recommended. The new umbrella DOI will be furnished as soon as possible. For now, we have included a placeholder statement in Section 5.*

*One additional note: it appears there was a PDF conversion issue, which truncated the DOI hyperlinks in Table 3, sending reviewers to an error landing page. That has been corrected.*

Line 95: "last" I think you mean 'most recent'?

*Indeed. We have removed this particular sentence and shifted much of the associated discussion to Section 6, where we discuss future scientific uses for the data collection.*

Line. 118: "new" I think you mean additional, especially because you have just expended several sentences to justify isotopic measurements based on past data.

*Substitution made.*

Section 2: not clearly specified, perhaps will clarify later, but I suspect:
7 vapor-phase measurements: 2 at BCO, 1 on ATR, 1 on P-3, 3 on ships, sum = 7? 5(?)
liquid-phase (precip) measurements: 1 at BCO, 3 on ships above, 1 on additional ship = 5? 3 seawater samplers on 3 ships = 3?

*Yes; we have rephrased here and in the Introduction to make this more clear.*

From Fig 2, rainwater samples occurred mostly along longitude approx 57W, with relatively few exceptions.

*The eye is drawn to that meridian because the Meteor, represented by magenta symbols, sampled almost exclusively there (over a relatively short N-S transect). However, as seen in Fig. 3, rain samples from the Meteor comprise but a small portion of the total number of rain samples.*

Line 145: very strong statement here: "… no island effects …". No upwind island effects? This needs referencing to back it up?

*The appropriate reference (Stevens et al., 2016) was cited in the previous sentence and has now been moved (one sentence down) for clarity.*

Line 148, "regionally representative": I think you mean representative across tropical trade wind environments globally but as written readers could interpret statement as referring only to local BCO environment? Needs some revision.

*Indeed. Revised as suggested.*

Line 198, "stored at room temperature": this reader doubts that authors could keep samples at steady temperature during long-range (BCO to Freiburg lab) long-duration transport and storage. Perhaps not important for isotopes? Need some justification here?

*We have modified the sentence so that it begins: "except when in transit". Shipping samples for laboratory analysis is common practice and not expected to fractionate the samples so long as they are capped tightly and/or wrapped in a sealant like parafilm. If caps and/or seals are inadequate in preventing evaporation, colder temperatures can be advantageous because they minimize isotopic fractionation. However, true leaks are likely to be apparent in liquid samples, regardless of the storage temperature. We have reported storage temperatures mainly for consistency and because there were some differences from one platform to the next. However, as now emphasized in Section 4.1.1, these differences have had no detectable effect on cross-platform comparisons.*

Line 219: that the Picarro instrument worked (acceptably?) during deployment nearly a decade earlier provides little help here about reliability, precision, accuracy?

*As described in both Sections 2 and 3, the ATR analyzer was calibrated extensively in the field and after the EUREC$^4$A deployment. All post-processing corrections were based on these calibration checks.*

*To avoid confusion, near Line 219 we have rephrased as follows: "Isotopic measurements aboard the ATR were made with a customized, fast-response version of Picarro's L2130-i cavity*

*ring-down spectrometer (with nominal sampling frequency of 1 Hz). The analyzer had been deployed in previous field campaigns, both for near-surface (Thurnherr et al., 2020) and airborne (Sodemann et al., 2017) measurements."*

Line 222: please give temperatures consistently in K or C, or explain why one seems appropriate in some cases but not others? Here readers confront inlet reference temperatures in K followed - within same sentence - by tubing heating temperatures in C! One or the other? Not both unless justified for valid reasons.

*Thank you for catching this. We have converted K to degrees C. Degrees C are already used exclusively in the Supplemental Information.*

Line 292: text here, e.g. "limit particle debris" bears (too) remarkable similarity to earlier text describing aircraft inlets (e.g. line 224). One suspects authors adopted text from ship and aircraft sources - fair enough - and no doubt particulate contamination proves troublesome in both cases but authors either need to acknowledge similarity of text (and sampling challenge) or take more care about repeating identical phrases.

*The same filter was used in both cases for the same purpose. However, we have modified the text in the first instance to avoid using the same exact turn of phrase: "Ambient air was pumped at a rate of 13 SLPM…through the gooseneck, past a particle filter, and down a 1.5 m long, 10 mm ID PTFE tube…"*

Line 294: "sniff tests"? Sloppy, at best. Sniff for RH changes? For odiferous tracers? Certainly not for isotopic composition! Again one suspects authors repeat text from ship-board operators / data providers but seems seriously out-of-place in a careful presentation of isotope data.

*We have replaced "sniff tests" with "empirical time-response tests".*

Line 427, Section 3 on data processing expends several sentences on water vapor quantification, both as mole fraction and as humidity. Surprised because not addressed in measurement sections?

*We have revised the top of Section 2 so that it is clear that all analyzers deployed during EUREC$^4$A measure water vapor as a mole fraction. The statements at the top of Section 3 are meant to explain that some datasets have converted mole fraction to another expression of humidity, such as specific humidity, for ease of comparison with other humidity sensors on the same platform.*

Line 439: "water vapor isotopic measurements varied widely". Okay, not really a surprise, but have authors given us tools to adjust expectations and make use? For this reader, no.

*To clarify, the statement here makes the case that the post-processing of measurements varied widely. In other words, bias corrections were tailored to each individual analyzer, as is best practice in the water vapor isotope measurement field. We have added new material to the Introduction to clarify this need upfront: "Uncertainties in these data reflect the diverse operating conditions and constraints associated with each platform and the need to tailor post-processing corrections to individual instrument performance, as is considered best practice (Aemisegger et al., 2012; Bailey et al., 2015)".*

*We have also made minor modifications to the top of Section 3.1, adding, for example, the following statements: "post-processing corrections ensure that measurements from any one platform are not only self-consistent but also comparable with measurements from other EUREC⁴A platforms or with isotopic data from previous field deployments…Accounting for…uncertainties and excluding data flagged for quality concerns will ensure that cross-platform comparisons are as accurate as possible". By applying the appropriate, tailored corrections, and flagging data of quality concern, we have made sure that the remaining high-quality data are comparable.*

Side-by-side systems at BCO should have provided "reference dataset with 1 minute time resolution". Failed. Authors conclude (line 793): "unexpected discrepancy between the BCO analyzers". Comparable systems on ATR and P-3 also failed intercomparison critera. Data not bad from ATR (albeit with uncertain response times and corrections) but not - unfortunately - comparable to P-3 data which, apparently, only reference to prior P-3 data. Comparable systems on ships? Only barely useful. Authors know these discrepancies better than any reader but fail to provide a coherent summary. If they can't extract useful summary, who can?

*These comments make us realize that "reference" is probably too loaded a term. We have replaced "reference" with "continuous fixed-point". Also, although one of the BCO analyzers exhibited problems with its spectroscopy and unexpected isotopic biases, the other analyzer's dataset serves as a downwind anchor point for ship-based and airborne measurements made to the east, which is already discussed both in Sections 4 (time series comparisons between the Meteor and BCO) and 6 (opportunities to carry out Lagrangian analyses).*

*Perhaps we have downplayed the very strong coherence in the EUREC⁴A isotopic data to a fault. For example, Fig. 13 demonstrates that the two airborne analyzers show very strong coherence in the boundary layer, where most of the ATR data are from. Similarly, Figs. 10-11 show very strong coherence in campaign-mean water vapor, rainwater, and seawater values. We have revised much of Section 4.1 so that the data consistency emerges as the key take-home message. We have not removed any discussion around cross-platform discrepancies; however,*

*we have attempted to communicate more clearly that these discrepancies are either the result of unique platform issues (e.g. spectroscopic oscillation in the BCO OA-ICOS time series), which we have identified and flagged, or the result of real environmental variability (e.g. lower isotope ratios associated with frontal rain as compared to shallow convective precipitation).*

Authors informed assessments of data utility (e.g. Section 4):
For vapor phase, perhaps a "subtle" (or, later, "very subtle) latitudinal pattern emerges. This reader can neither see nor credit such spatial patterns but accepts that authors and other users might. These cautions need to move earlier, e.g. in abstract?

*Perhaps we've misunderstood the criticism, but, in our opinion, the subtle patterns indicate strong coherence, which (as pointed out near the top of Section 4.1.1, lends "confidence to the measurement accuracy"). We have revised Section 4.1.1 such that the take home messages are, in order of importance, as follows:*

1. *Data coherence is strong, indicating high quality data*
2. *Accounting for quality flags is important for accurate cross-platform comparisons*
3. *Differences in high-quality data represent real environmental variability and are consistent with theoretical expectations.*

*The specific reference to the "very subtle" pattern in rainwater samples has been removed, as we choose to emphasize, instead, how very close in mean value the rainwater samples are.*

Storage effects, which vary from 'frozen' to even 'poisoned then frozen' to room temperature and even 'subject to drastic heating', impose evaporation and biological effects on isotope ratios. Authors recognize such issues and provide (where possible) summaries of storage protocols but only rarely deal with the larger issue? E.g. Brown samples should differ substantially from Meteor samples due to differences in storage? Systematic or erratic? Not clear and not well addressed?

*As explained in a previous response, in general, we do not expect basic storage differences among platforms to affect sample quality. We have also added a statement to Sect. 4.1.1 that suggests that the high cross-platform consistency provides compelling evidence that differences in sample storage did not influence sample isotope ratios.*

*As now more clearly emphasized in the Introduction and Sect. 4, we do not recommend liquid water samples flagged for data quality concerns for most scientific analyses.*

Line 1425: Given the scale of aircraft tracks, land mass (grey) in Figure 2 represents South America with Trinidad/Tobago clearly visible. Barbados (13N, 59W), apparently in black, fails to

appear to these old eyes even under extreme zoom. All land masses should stay dark grey, particularly if you want color matching to Fig 3. Show location of Deebles Pt BCO? As readers find later, this scale driven by satellite products while less helpful for immediate locale.

*Actually, the maps are scaled to fit the in-situ data. The EUREC$^4$A satellite datasets cover a much larger region than that shown.*

*As per the suggestion, we have remade the Figures so that Barbados appear in gray, rather than black.*

Authors make extensive reference to and use of column-integrated satellite products, on two or more spatial scales. Not surprising given EUREC4A relevance and motivation, and authors have offered good access and reasonable interpretation. Summary (in my words): not bad, nothing in remotely sensed data proves or disproves EUREC4A in situ data but subsequent users should take great care in any such intercomparisons for a variety of reasons. Some such caution should emerge as a clearer outcome?

*The idea of including the EUREC$^4$A remotely sensed isotope ratios was to provide broader spatial and temporal context for the in-situ measurements with the full understanding that such comparisons require care. Indeed, Section 4.2 contains several strong cautionary statements about comparing remotely sensed and in-situ measurements. Note, for example, the following: "As demonstrated in Fig. 14, the satellites provide rich spatial context for the in-situ data. Nevertheless, when using the two in tandem, care must be taken to consider differences in what each type of measurement represents." The next several paragraphs all discuss the different sensitivities of the measurements and the need to consider averaging kernels.*

*To provide more caution up front, we have added statements to the end of the Introduction and to the beginning of Section 4.2 that emphasize the vastly different sensitivities to the atmosphere between the remote sensors and the water vapor isotopic analyzers deployed in situ.*

Because of focus on vertical profiles driven by remote sensing, this reader notes again absence of radiosonde profiles (e.g. BCO must launch sondes daily) and of dropsonde profiles from HALO and P-3. Because EUREC4A expended efforts in track planning and resources in sondes themselves, and because correlation with water vapor / humidity turns out such an important factor in isotope measurements, why have authors not at least mentioned sonde humidity profiles? Even to say 'not useful'. To this readers, seems a strange omission.

*Based on this recommendation, we now encourage data users to seek out ESSD special issue papers on EUREC$^4$A radiosonde and dropsonde measurements at the top of Section 4 and explain at the top of Section 3 that multiple expressions of water vapor concentration are often*

*included in the data files for ease of comparison with other sensors on the same platform, dropsondes, or radiosondes.*

*Sensor intercomparisons have also been performed in other already published ESSD papers; therefore, we decided not to repeat these results in this manuscript. To satisfy the editor's well-appreciated curiosity, we have reproduced Fig. 16 from Bony et al. (2022) below, which compares specific humidity data (in g/kg) from different sensors onboard the ATR, including the Picarro isotopic analyzer (teal), and the HALO dropsonde data for different flight patterns ("R", "L", "S").*

[Figure]

Authors use term 'diurnal' when in fact they mean 'diel'. Strictly, diurnal refers to daylight, nocturnal refers to night, diel refers to full 24-hour cycle.

*Fixed.*